# Metabolic profiling of *Mytilus coruscus* mantle in response of shell repairing under acute acidification

**Xiaojun Fan**[1☯], **Ying Wang**[1☯], **Changsheng Tang**[1], **Xiaolin Zhang**[1], **Jianyu He**[1], **Isabella Buttino**[2], **Xiaojun Yan**[1]*, **Zhi Liao**[1]*

**1** Laboratory of Marine Biology Protein Engineering, Marine Science and Technical College, Zhejiang Ocean University, Zhoushan City, Zhejiang, China, **2** Italian Institute for Environmental Protection and Research (ISPRA), Rome, Italy

☯ These authors contributed equally to this work.
* liaozhi@zjou.edu.cn (ZL); yanxj@zjou.edu.cn (XY)

**Data Availability Statement:** All relevant data are within the manuscript and its Supporting information files. All raw mass spectrometry data for metabolomic analysis are available from the

## Abstract

*Mytilus coruscus* is an economically important marine bivalve mollusk found in the Yangtze River estuary, which experiences dramatic pH fluctuations due to seasonal freshwater input and suffer from shell fracture or injury in the natural environment. In this study, we used intact-shell and damaged-shell *M. coruscus* and performed metabolomic analysis, free amino acids analysis, calcium-positive staining, and intracellular calcium level tests in the mantle to investigate whether the mantle-specific metabolites can be induced by acute seawater acidification and understand how the mantle responds to acute acidification during the shell repair process. We observed that both shell damage and acute acidification induced alterations in phospholipids, amino acids, nucleotides, organic acids, benzenoids, and their analogs and derivatives. Glycylproline, spicamycin, and 2-aminoheptanoic acid (2-AHA) are explicitly induced by shell damage. Betaine, aspartate, and oxidized glutathione are specifically induced by acute acidification. Our results show different metabolic patterns in the mussel mantle in response to different stressors, which can help elucidate the shell repair process under ocean acidification. furthermore, metabolic processes related to energy supply, cell function, signal transduction, and amino acid synthesis are disturbed by shell damage and/or acute acidification, indicating that both shell damage and acute acidification increased energy consumption, and disturb phospholipid synthesis, osmotic regulation, and redox balance. Free amino acid analysis and enzymatic activity assays partially confirmed our findings, highlighting the adaptation of *M. coruscus* to dramatic pH fluctuations in the Yangtze River estuary.

## Introduction

Since the end of the last century, biologists have raised concerns about how ocean acidification (OA) affects calcification in corals, a foundation marine taxon [1]. Over the last decade, the OA research community has highlighted the vulnerability of calcification-dependent marine organisms to OA [2]. Queirós, *et al.* [3] reported that, $CO_2$ levels in the global atmosphere

MetaboLights database with the Accession Number MTBLS5530.

**Funding:** This study was supported by the National Natural Science Foundation of China (Grant Nos. 32271580, 42020104009, and 32200083). Zhejiang Provincial Natural Science Foundation (Grant Nos. LQ23D060002 and LTGS23C010001). The funders had no role in study design, data collection and analysis, decision to publish, or preparation of the manuscript.

**Competing interests:** The authors have declared that no competing interests exist.

have increased from ~280 ppm (pre-industrial) to ~400 ppm today. According to the climate change model, Richard *et al.* [4] predicted that OA may last a few centuries, and the ocean water pH value will be 7.4 in the year 2300. The increasing atmospheric $CO_2$ levels changed seawater carbon chemistry and have acidified oceans [5]. Although some species are resistant to OA [6], others are vulnerable and have far-reaching implications for oceanic ecosystems [7]. Therefore, OA has attracted considerable attention worldwide.

Bivalves are an important source of dietary animal protein [8], and mollusk aquaculture production contributes to 21.42% of the total aquaculture production worldwide [9]. Furthermore, bivalves play critical ecological roles in aquatic environments, including habitat creation [10], suspension of organic waste filtration [11], shoreline protection [12], and controlling the abundance of harmful algal species [13]. The deleterious effects of OA on bivalve' biological processes have been frequently reported in recent years, including decreased immune capacity [14], growth and development retardation [15], loss of shell structural integrity [16], and reduction in byssus strength [17]. Among the adverse effects of OA on bivalves, shell biomineralization has received increasing attention [18, 19]. Calcified shells are important for bivalves because they protect them from tides, predators, and other harsh environmental factors [20]. The shell calcification process of bivalves is strongly influenced by the pH of water [21, 22]. Consequently, under acidic conditions, many bivalve show compromised shell growth and integrity, and an increased vulnerability to diseases and parasites, resulting in higher mortality [23, 24]. Notably, *Mytilus* somehow presented a resistance to OA. Thomsen, *et al.* [25] reported that, *Mytilus edulis* tolerates high ambient $pCO_2$, and its growth and calcification rates were seemingly unaffected by OA. Although food supply and energy availability have been suggested to be important factors in the tolerance of *M. edulis* to OA [25]. the molecular mechanism underlying this tolerance is still unknown. Previous studies have revealed significant short-term selective responses of traits directly affected by OA and long-term adaptation potential in *Mytilus* [26]. Considering that previous studies focused on the medium- or long-term effects of OA on mussels [27], it is necessary to explore the rapid response of mussels to OA.

Bivalve shells are continuously exposed to different stressors in the natural environment, such as predators and parasites like worm [28] and borers [29], and anthropogenic activities like dredging [30], which can lead to shell fracture and injury. In addition, *Mytilus* shells are frequently damaged by collisions and friction between individuals during the high-density farming process or between the mussels and reef under wave-hurled projectiles [31]. Therefore, *Mytilus* and other bivalves may have evolved shell repair mechanisms to counter shell injuries. Shell repair processes in bivalve molluscs have been studied in some species [28, 32, 33], and common mechanisms have been identified in *Mytilus* and other bivalves, including the formation of a periostracum-like layer in the damaged shell area, calcite secretion onto the organic layer, and shell formation [34]. This process is analogous to the normal biomineralization process in bivalves, and the edge region of the mantle plays a key role in the shell repair process [33, 34]. Furthermore, the estuary sea area, where *Mytilus coruscus* lives, shows seasonal fluctuations in natural pH due to freshwater input; thus, the seawater pH value is rapidly reduced in summer [35], indicating that *M. coruscus* would be affected if it cannot rapidly repair the damaged shell and recover from short-term exposure to coastal acidification. Previous studies have shown that environmental factors inhibit the shell damage-repair process in bivalves. For example, increased temperature hinders the ability of shell repair in shell-damaged *Mya truncate* [36]. However, the shell repair process under acidic conditions in bivalves is largely unknown. Metabolomics techniques can effectively demonstrate the physiological response of marine organisms to OA [37]. Therefore, in this study, we used *M. coruscus* to analyze the metabolism of the mantle tissue between intact-shell and damaged-shell mussels

under normal seawater (pH 8.1 for 48 h) and acute acidified seawater (pH 7.4 for 48 h). The study aimed to: 1) reveal the metabolic response and possible adaptation mechanism of *Mytilus* in acidified seawater, and 2) understand how the mussel mantle implements a shell-repair process in acidified seawater.

## Materials and methods

### Ethics statement

All procedures were performed in accordance with the guidelines of the Regulations for the Administration of Laboratory Animals (Decree No. 2 of the State Science and Technology Commission of the People's Republic of China, November 14, 1988) and was approved by the Institutional Animal Care and Use Committee of Zhejiang Ocean University.

### *Mytilus coruscus* sampling and treatment

*M. coruscus* adult individuals were collected from a mussel farm located in the Shengsi Sea area of Zhoushan Islands. The mussels were acclimatized in a tank with aerated natural seawater (22 ˚C, salinity 25 ppt, pH 8.1) for 7 days. After acclimatization, the mussels were randomly divided into two groups, intact-shell and damaged-shell mussels. The damaged-shell mussels were prepared using the shell-drilling methods as described by Yarra, *et al* [33]. Each group was subdivided into two subgroups and raised for 48 h in tanks with normal sea water (pH 8.1) and acidified sea water (pH 7.4), respectively. The pH of the sea water was monitored using a pH meter, and precisely controlled using a Seawater Acidifier (Starfish SF0S02, Qingdao, China) with a $CO_2$ pump. Three pH treatment-level replications were used for the mussels in this study. For each pH treatment-level replication, the mussels with intact-shell and damaged-shell were mixed and raised in the same tank. Six tanks were prepared for the mussels and 30 mussel individuals, including 15 intact-shell and 15 damaged-shell mussels were raised in each tank. A total of 180 mussels were used for our experiment, and 45 individuals for each group. In summary, four mussel groups were prepared for the following studies: mussels with intact shell fed in normal sea water (designated as the CN group), mussels with intact shell fed in acidified sea water (designated as the CA group), shell-drilled mussels fed in normal sea water (designated as the DN group), and shell-drilled mussels fed in acidified sea water (designated as the DA group) (Fig 1).

### Mantle metabolites extraction

After treatment, the mantle tissues were dissected and removed from the mussels from each group. The mantle edge was collected, and the metabolites were extracted using the protocol described in a previous work [38]. Briefly, L-2-chlorophenylalanine (0.3 mg/mL in methanol) was used as the internal standard, and methanol (80% in water) was used as the extraction solvent for each sample. The extracts were centrifuged at 4˚C (11,000 × g) for 10 min after grinding with steel balls at 60 Hz for 2 min. The supernatant was lyophilized and dissolved in methanol (20% in water). After centrifugation (4 ˚C, 11,000 × g, 10 min), the supernatant was collected using crystal syringes, filtered through 0.22 μm microfilters, and sent for LC-MS analysis. Quality Control (QC) samples were prepared by mixing aliquots of all samples to form a pooled sample.

### UPLC-MS/MS analysis

An ACQUITY UPLC I-Class plus system (Waters, Milford, CT, USA) coupled with a Q-Exactive plus Orbitrap mass spectrometer (MS) with electrospray ionization (ESI) source (Thermo

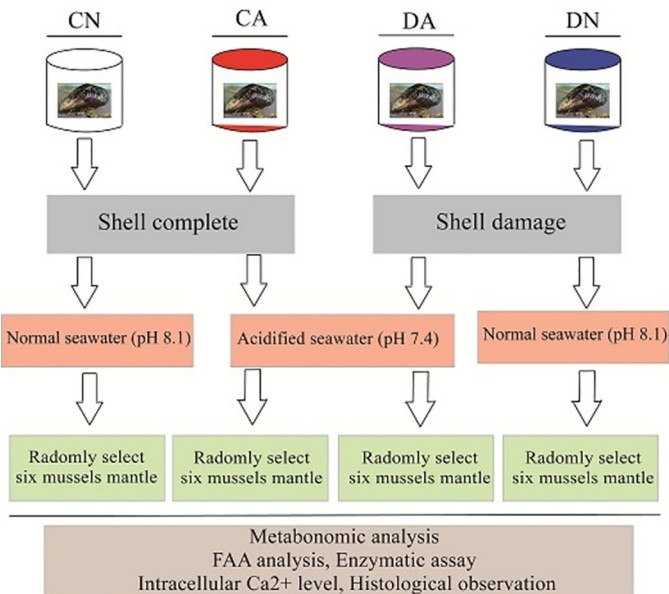

**Fig 1. Schematic view of the experimental design.**

Fisher Scientific, Waltham, MA, USA) was used to analyze the metabolic profiles in both ESI positive and negative ion modes. A HSS T3 column (1.8 μm, 100×2.1 mm) was used for metabolic isolation. Water and Acetonitrile/Methanol of 2/3 (v/v) containing 0.1% formic acid were used as mobile phases A and B, respectively. The proportion of solution B increased from 5% to 100% within 15 min of elution. The flow rate was 0.35 mL/min and the column temperature was set as 45 ˚C.

For MS analysis, the capillary temperature and the gas heater temperature were set as 320 ˚C and 350 ˚C, respectively. The spray voltage was set to 3,800 V for the positive ion model and -3,000 V for the negative ion model. Argon (99.999% purity) was used as the collision-induced dissociation gas. Low-energy (4eV) and high-energy (20-45eV) scan were used to fragment ions.

Data acquisition was performed in the full scan mode, with *m/z* ranging from 100 to 1,200 in the MSE mode. The scan time and the interscan delay were set as 0.2 s and 0.02 s, respectively. The full MS and the MS/MS resolutions were set to 70,000 and 17,500, respectively. QC samples were injected at regular intervals throughout the analytical run to provide a set of data from which repeatability could be assessed.

## Data preprocessing and multivariate statistical analysis

Raw UPLC-MS data were processed and analyzed using Progenesis QI V2.3 (Waters Corporation, Milford, USA) using the following parameters: precursor and product tolerance were set at 5 ppm and 10 ppm, respectively, and retention time (RT) tolerance was set at 0.02 min. Metabolites with relative standard deviation (RSD) > 30% in the QC samples were excluded. The extracted data were further processed by removing peaks with missing values (ion intensity = 0) in more than 50% of the samples. Three-dimensional datasets, including m/z, peak RT, peak intensities, and RT–m/z pairs, were used as identifiers for each ion. Metabolites were identified by Progenesis QI using a precise mass-to-charge ratio (m/z), secondary fragments, and isotopic distribution against public databases, including the Human Metabolome

Database (HMDB, http://www.hmdb.ca/), Lipidmaps (http://www.lipidmaps.org.), and METLIN (http://metlin.scripps.edu/).

A data matrix was created from the positive and negative ion data and imported into the SIMCA software package (version 14.0, Umetrics, Umeå, Sweden). Principal Component Analysis (PCA) and Orthogonal Partial Least-Squares-Discriminant Analysis (OPLS-DA) were performed to visualize the metabolic profiles of the four groups. To prevent overfitting, 7-fold cross-validation and 200 Response Permutation Testing (RPT) were used to evaluate the quality of the model.

The variable Importance of Projection (VIP) values obtained from the OPLS-DA model were used to rank the overall contribution of each variable to group discrimination. A two-tailed Student's *t*-test was performed to verify whether the metabolite differences between the groups were significant. Differential metabolites were selected with VIP values more than 1.0 and *P* values less than 0.05.

Quantitatively different metabolites were mapped to the reference pathway using the Kyoto Encyclopedia of Genes and Genomes (KEGG; http://www.genome.jp/kegg/) database. Significantly enriched pathways were assessed based on false discovery rate-adjusted hypergeometric test statistics ($P < 0.05$).

## Free amino acid (FAA) analysis of the mantle sample

The mantle tissues used for metabolic analysis were further processed and subjected to FAA analysis using an Amino Acid Analyzer (LA8080, Hitachi, Japan) following the protocol described in our previous study [39]. Briefly, the dissected mantle edge sample was grounded into powder in liquid nitrogen, homogenized in deionized water, and then ultra-sonicated and centrifuged (4 ˚C, 8,000 ×g, 20 min). The supernatant was collected and ultra-filtrated with 3 kDa cut-off, and the filtered solution was collected, filtrated using a 0.22 μm membrane, and freeze-dried. The freeze-drying powder was diluted with hydrochloric acid and separated by an aluminum ion exchange column (4.6×60.0 mm, Hitachi, Japan) and analyzed by the Amino Acid Analyzer, using the standard amino acid mixture (AN-II, Hitachi, Japan) for FAA qualitative and quantitative analysis.

## Enzymatic assay

Mantle tissue was collected from the mussels using the same treatment described above. After grinding in liquid nitrogen, followed by homogenization and centrifugation, the supernatant of the mantle sample was quantified for total protein and enzyme activity. The total protein concentration was determined using the bicinchoninic acid standard method [40]. The activities of superoxide dismutase (SOD), catalase (CAT), glutathione (GSH), 5'-nucleotidase(5'-NT), total antioxidant capacity (T-AOC), and total nitric oxide synthase (T-NOS) were measured using the commercial kits (Jian-Cheng Bioengineering Research Institute, Nanjing, China) according to the manufacturer's instruction.

## Intracellular Ca$^{2+}$ level of the mantle

Ca$^{2+}$ levels were determined based on the method described by Wang *et al*. [41], with slight modifications. Mantle edge samples were dissected, homogenized, and incubated with collagenase (2.5 mg/mL) in Hank's buffer without Ca$^{2+}$ or Mg$^{2+}$. The mixture was stirred gently at 25 ˚C for 20 min. The intracellular Ca$^{2+}$ concentration at the mantle edge was measured using Fluo-3 method with a Ca$^{2+}$ concentration detection kit (Solarbio, Beijing, China) according to the manufacturer's instruction. The median fluorescence intensity of 10,000 cells was recorded

as relative $Ca^{2+}$ levels using a flow cytometer (CytoFLEX, Beckman Coulter, Brea, CA, USA) and CytExpert 2.5 software.

### Histological observation of the mantle edge

Mantle edges from the four mussel groups (CN, DN, CA and DA) were dissected and immediately fixed in 4% paraformaldehyde for histological observation. Tissue samples were dehydrated, embedded in paraffin, and sectioned at 4 μm using a microtome (HistoCore BIOCUT, Leica Biosystems, Deer Park, IL, USA). The localization of $Ca^{2+}$ in the mantle tissue was determined using the Alizarin Red S (ARS) staining method according to a previously described protocol [42]. The distribution of insoluble calcium carbonate in the mantle tissue was determined using the von Kossa staining method, according to a previously described protocol [43]. Stained sections were examined under a light microscope (ECLIPSE E100, Nikon, Tokyo, Japan).

## Results

### Overall changes in metabolites identified from the mantle tissue

A total of 17,366 ion peaks, including 6,779 from the positive model and 10,587 from the negative model, were detected in the mantle samples after removing low-quality ions with a relative standard deviation (RSD)>30%. A total of 7,882 metabolites were identified, 3,398 from the positive model and 4,484 from the negative model. Ion chromatograms from the four QC samples showed no distinct peak drifts with stable retention times for both the positive and negative models (S1 Fig), indicating good instrument stability. We compared the metabolic profiles between different groups (CN, CA, DN, and DA), and the metabolite intensity distributions for the four sample groups and QC samples are shown in S2 Fig, indicating the stability of the QC samples and the difference in metabolite intensity among the tested samples. All raw mass spectrometry data for metabolomic analysis were uploaded to the MetaboLights database with the Accession Number MTBLS5530.

The changes in the metabolite profiles of the four mantle sample groups were detected using two-dimensional PCA score plots. As shown in Fig 2, samples from different groups showed separation with a few intermixed samples. However, the OPLS-DA models showed significant separation for the five pair-wise comparisons of the four sample groups (Fig 3), indicating that shell damage and/or acute acidification significantly changed the metabolite profile of the mantle.

The interpretation abilities of the model were 0.977, 0.992, 0.998, 0.994, and 0.994, and the predictive abilities were 0.615, 0.859, 0.917, 0.865, and 0.656 for the five pair-wise comparisons (DN *vs*. CN, CA *vs*. CN, DA *vs*. CN, DA *vs*. DN, and DA *vs*. CA), respectively (S1 Table and S3 Fig). These results indicated that the OPLS-DA model did not exceed the fitting. In conclusion, the OPLS-DA model showed good interpretation and prediction abilities, reflecting the difference between the groups.

### Screening of the significantly differential metabolites (SDMs)

Metabolites with VIP >1 and $P < 0.05$ in different pair-wise comparisons were selected as significantly differential metabolites (SDMs) among the four sample groups. The number of SDMs from each pair-wise comparison is summarized in Fig 4, and volcano maps for visualizing the $P$ value, VIP value, and fold change (FC) of the SDMs in different pair-wise comparisons are summarized in S4 Fig.

The SDMs were identified by searching against the HMDB, lipidmap, and METLIN databases. The top 50 SDMs with the highest VIP values in the five pair-wise comparisons are listed

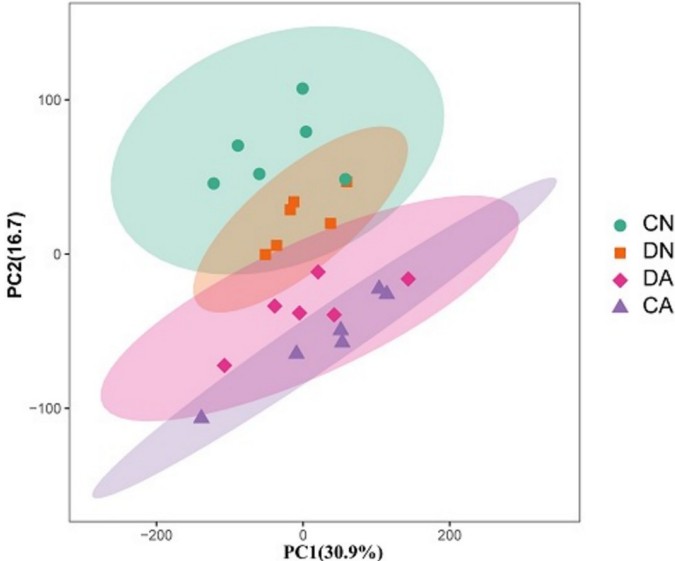

**Fig 2. Principal component analysis (PCA) of the metabolites at different phases for the twenty-four mantle samples.** CN, the mussel with complete shell and fed in normal sea water (pH 8.1); DN, the mussel with drilled shell and fed in normal sea water (pH 8.1); CA, the mussel with complete shell and fed in acidified sea water (pH 7.4) with exposure time of 48 h; DA, the mussel with drilled shell and fed in acidified sea water (pH 7.4) with exposure time of 48 h.

in Fig 5. The identified SDMs were further annotated against the KEGG database, and 74, 121, 123, 103, and 36 SDMs were annotated with KEGG ID for DN *vs.* CN, CA *vs.* CN, DA *vs.* CN, DA *vs.* DN, and DA *vs.* CA, respectively.

## Characterization of key metabolic pathways enriched by SDMs

To further understand the function of SDMs, the KEGG ID of annotated SDMs was used for enrichment analysis to identify the pathways enriched in SDMs. The enriched pathways (*P* value < 0.05) are shown in Fig 6. We observed that in DN *vs.* CN comparison, 30 KEGG-annotated SDMs were enriched in 10 KEGG pathways with *P* values < 0.05, and purine metabolism, glycerophospholipid metabolism, and mTOR signaling pathway were the three KEGG pathways with the lowest *P* values (Fig 6A). In the CA *vs.* CN comparison, 57 SDMs were significantly enriched in 13 KEGG pathways, and purine metabolism, aminoacyl-tRNA biosynthesis, and D-Arginine and D-ornithine metabolism were the top three KEGG pathways (Fig 6B). In DA *vs.* CN, 52 SDMs were significantly enriched in 12 KEGG pathways, and purine metabolism, aminoacyl-tRNA biosynthesis, and mTOR signaling pathway were the top three KEGG pathways (Fig 6D). In DA *vs.* DN, 47 SDMs were significantly enriched in 11 KEGG pathways, and Aminoacyl-tRNA biosynthesis, ABC transporters, and purine metabolism were the top three KEGG pathways (Fig 6D). In DA *vs.* CA, only two KEGG pathways, pyruvate metabolism and glycerophospholipid metabolism, were significantly enriched in four SDMs (Fig 6E).

## FAA analysis

FAA content was determined using an amino acid analyzer for mantle samples of the CN, DN, CA, and DA groups. The standard curve used for qualitative and quantitative analysis of FAA and other nitrogenous compound was shown in S5 Fig. Sixteen standard amino acids were

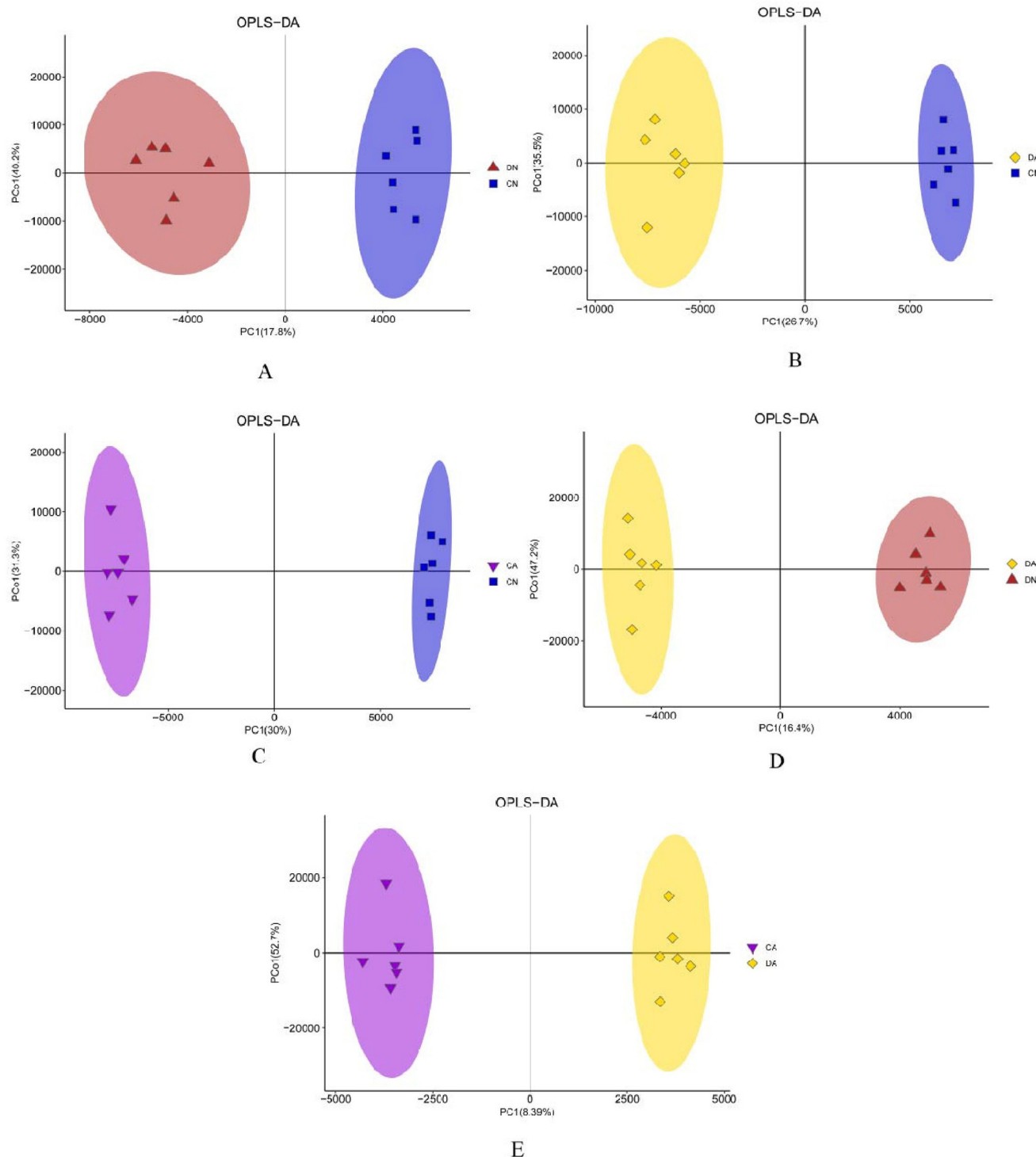

**Fig 3. Orthogonal projections to latent structures discriminant analyses (OPLS-DA) score plots derived from the LC-MS profiles of the mantle samples.** A: OPLS-DA score plot by DN *vs*. CN; B: OPLS-DA score plot by DA *vs*. CN; C: OPLS-DA score plot by CA *vs*. CN; D: OPLS-DA score plot by DA *vs*. DN; E: OPLS-DA score plot by CA *vs*. DA. CN, the mussel with complete shell and fed in normal sea water (pH 8.1); DN, the mussel with drilled shell and fed in normal sea water (pH 8.1); CA, the mussel with complete shell and fed in acidified sea water (pH 7.4) with exposure time of 48 h; DA, the mussel with drilled shell and fed in acidified sea water (pH 7.4) with exposure time of 48 h.

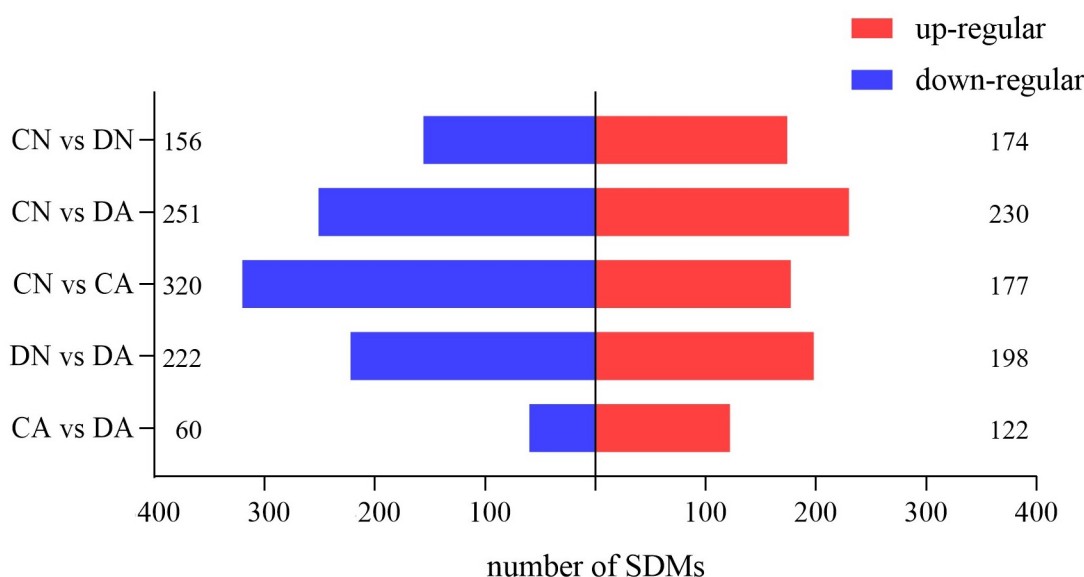

**Fig 4. Number of SDMs (VIP > 1, *P* < 0.05) from five pair-wise comparisons among the mantle samples.** CN, the mussel with complete shell and fed in normal sea water (pH 8.1); DN, the mussel with drilled shell and fed in normal sea water (pH 8.1); CA, the mussel with complete shell and fed in acidified sea water (pH 7.4) with exposure time of 48 h; DA, the mussel with drilled shell and fed in acidified sea water (pH 7.4) with exposure time of 48 h.

detected in the mantle samples (Table 1), in which, Asp, Gly, and Arg were the most abundant FAAs with a concentration of more than 100 μg/g of dry tissue. Met, Ile, and Leu showed the lowest concentrations of less than 10 μg/g of dry tissue. In addition, some amino acid derivatives and nitrogen-containing components were also detected, including taurine (Tau), urea, anserine (Ans), β-alanine (β-Ala), 5-hydroxylysine (Hylys), and ornithine (Orn) (Table 1). We observed that compared with the CN group, the concentrations of Glu and Arg were significantly up-regulated in both the CA and DA groups. Leu and urea were significantly up-regulated in the CA group, and ornithine was significantly up-regulated in the DA group (Table 1).

## Enzyme activity and intracellular Ca$^{2+}$ level of the mantle tissue

The enzyme activities of T-AOC, 5'-NT, CAT, T-NOS, and SOD, and GSH levels were determined for the mantle tissue. As shown in Table 2, compared to the CN group, shell damage (DN group) showed significantly lower CAT activity and higher GSH levels. Acute acidification (CA group) significantly increased CAT and SOD activities and increased the GSH levels in the mantle. Moreover, the combined shell damage and acute acidification (DA group) decreased the activity of T-AOC and SOD, and increased CAT activity and GSH levels.

Intracellular Ca$^{2+}$ levels in the mantle tissue were determined using Fluo-3 fluorescent probe and counted using a flow cytometer. The fluorescence intensity per 10,000 cells for mantle samples from the four groups is summarized in Fig 7, and the corresponding histograms from the flow cytometer are listed in S6 Fig. We noted that the intracellular Ca$^{2+}$ level was up-regulated in the DN group and down-regulated in the DA group; however, this difference was not significant compared with that in the CN group. In addition, for the shell damaged mussels, acute acidification (DA group) significantly decreased the intracellular Ca$^{2+}$ level in the mantle compared to the DN group, indicating a suppressive effect of the shell repair process in response to acute acidification (Fig 7).

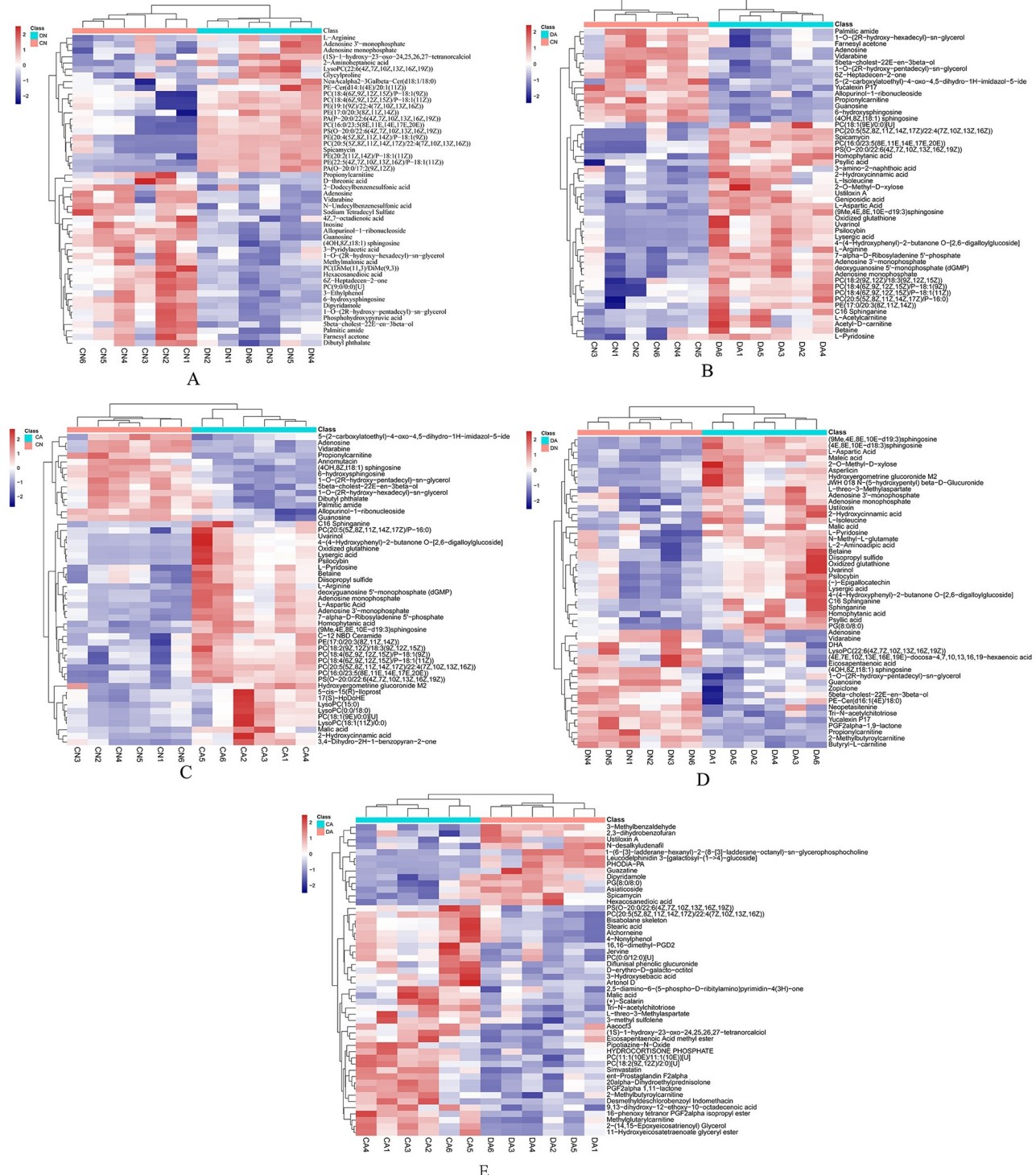

**Fig 5. Hierarchical clustering analysis of 50 SDMs with the highest VIP-value from five pair-wise comparisons.** A: Plot of hierarchical clustering analysis of 50 SDMs from DN *vs*. CN; B: Plot of hierarchical clustering analysis of 50 SDMs from DA *vs*. CN; C: Plot of hierarchical clustering analysis of 50 SDMs from CA *vs*. CN; D: Plot of hierarchical clustering analysis of 50 SDMs from DA *vs*. DN; E: Plot of hierarchical clustering analysis of 50 SDMs from CA *vs*. DA. CN, the mussel with complete shell and fed in normal sea water (pH 8.1); DN, the mussel with drilled shell and fed in normal sea water (pH 8.1); CA, the mussel with complete shell and fed in acidified sea water (pH 7.4) with exposure time of 48 h; DA, the mussel with drilled shell and fed in acidified sea water (pH 7.4) with exposure time of 48 h.

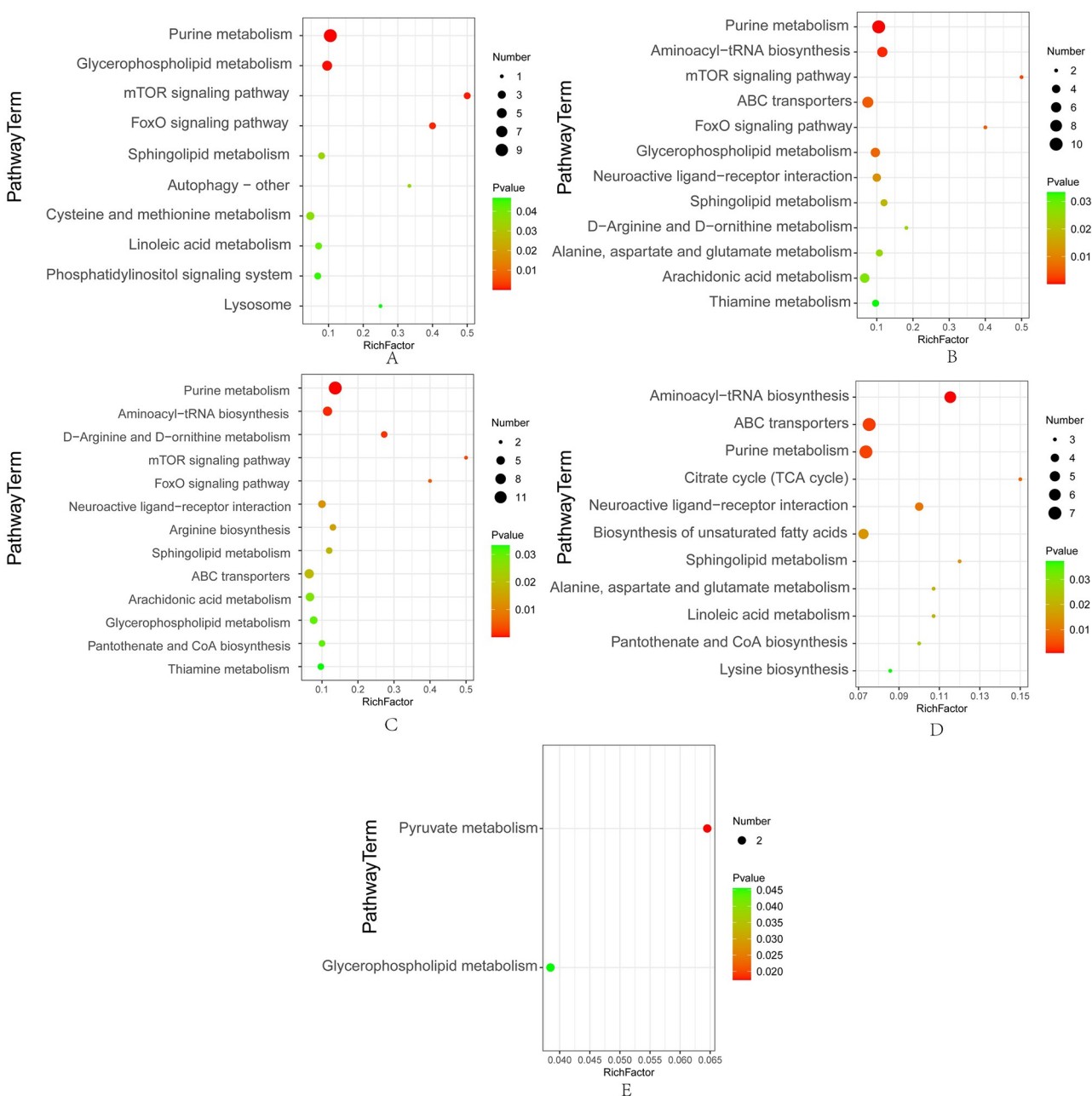

**Fig 6. Significantly enriched KEGG pathways for the differential metabolites from five pair-wise comparisons.** A: Significantly enriched KEGG pathways from the DN *vs.* CN; B: Significantly enriched KEGG pathways from the DA *vs.* CN; C: Significantly enriched KEGG pathways from the CA *vs.* CN; D: Significantly enriched KEGG pathways from the DA *vs.* DN; E: Significantly enriched KEGG pathways from the CA *vs.* DA. CN, the mussel with complete shell and fed in normal sea water (pH 8.1); DN, the mussel with drilled shell and fed in normal sea water (pH 8.1); CA, the mussel with complete shell and fed in acidified sea water (pH 7.4) with exposure time of 48 h; DA, the mussel with drilled shell and fed in acidified sea water (pH 7.4) with exposure time of 48 h.

## Histological observations

Histological observations were performed using ARS and von Kossa staining to show the localization of $Ca^{2+}$ and insoluble calcium carbonate, respectively, at the mantle edge. The mantle edge, particularly the outer mantle edge epithelial layer, has been implicated in bivalve shell

**Table 1. Free amino acid composition (μg/g of dry tissue) in the mantle of *M. coruscus* in different groups.**

| (μg/g of dry tissue) | CN | DN | CA | DA |
|---|---|---|---|---|
| Amino acid | | | | |
| Asp | 238.81±17.27 | 214.97±21.31 | 254.90±16.33 | 226.15±19.50 |
| Thr | 19.04±6.43 | 17.30±5.15 | 21.05±6.31 | 23.53±3.35 |
| Ser | 26.24±9.01 | 27.86±7.21 | 32.68±6.73 | 37.71±7.99 |
| Glu | 78.56±2.44 | 95.68±14.32 | 97.94±7.30 * | 113.19±25.41 |
| Gly | 168.11±35.16 | 159.74±7.28 | 209.62±25.76 * | 205.85±11.90 * |
| Ala | 59.90±20.01 | 67.86±8.84 | 72.40±9.26 | 72.99±13.26 |
| Val | 12.01±4.58 | 14.42±4.87 | 16.86±2.19 | 12.07±1.64 |
| Cys | 16.42±3.30 | 17.57±5.27 | 17.96±2.68 | 20.29±7.15 |
| Met | 7.46±2.25 | 6.36±1.61 | 8.00 ±2.13 | 7.44±1.94 |
| Ile | 7.49±1.61 | 7.34±1.08 | 8.94 ±1.36 | 8.61±0.63 |
| Leu | 6.51±1.40 | 6.76±1.03 | 7.86 ±1.68 | 7.44±1.32 |
| Tyr | 13.87±3.13 | 14.62±2.87 | 17.95 ±2.89 | 17.08±1.81 |
| Phe | 17.49±0.49 | 17.81±1.83 | 14.99 ±8.73 | 22.14±5.31 |
| Lys | 17.69±2.36 | 18.13±3.25 | 19.91±2.76 | 23.89±3.14 |
| His | 15.82±6.09 | 13.60±0.99 | 15.28±4.06 | 16.31±0.78 |
| Arg | 112.29±13.12 | 113.03±14.55 | 131.66±5.26 * | 139.93±2.41* |
| Nitrogenous compound | CN | DN | CA | DA |
| Tau | 1604.00±108.87 | 1557.09±135.02 | 1745.64±75.80 | 1681.35±47.20 |
| Urea | 678.40±26.17 | 707.81±21.45 | 795.54±18.74* | 648.37±34.96 |
| Ans | 66.11±14.18 | 69.34±4.46 | 73.21±5.70 | 79.38±5.31 |
| Orn | 12.08±2.26 | 11.77±3.27 | 13.81±2.94 | 14.55±2.19 * |
| β-Ala | 9.98±2.24 | 9.17±2.41 | 11.25±1.66 | 11.17±3.01 |
| Hylys | 7.46±0.55 | 7.13±0.65 | 6.95±0.25 | 6.85±0.36 |

Values are represented as means ± S.D (n = 3), and the * denotes significant difference ($P<0.05$) comparing with the CN group. CN, the mussel with complete shell and fed in normal sea water (pH 8.1); DN, the mussel with drilled shell and fed in normal sea water (pH 8.1); CA, the mussel with complete shell and fed in acidified sea water (pH 7.4) with exposure time of 48 h; DA, the mussel with drilled shell and fed in acidified sea water (pH 7.4) with exposure time of 48 h.

formation [44, 45]. In our study, the mantle edge of *M. coruscus* was composed of three folds, the outer fold (OF), middle fold (MF), and inner fold (IF) (Figs 8 and 9), representing three different regions of the mantle edge. The outer fold was similar to the middle fold in terms of length and shape, whereas the inner folds were apparently different. A columnar epidermal

**Table 2. Enzymatic activity of *M. coruscus* mantle in different groups.**

| Enzyme | CN | DN | CA | DA |
|---|---|---|---|---|
| T-AOC (U/mg prot) | 0.122±0.009 | 0.104±0.010 | 0.124±0.006 | 0.097±0.005* |
| 5'-NT (U/mg prot) | 0.125±0.024 | 0.136±0.086 | 0.127±0.033 | 0.100±0.265 |
| CAT (U/mg prot) | 49.558±2.304 | 31.489±6.930* | 71.895±10.053* | 56.899±20.042 |
| GSH (mg/mg prot) | 0.036±0.001 | 0.045±0.002* | 0.048±0.006* | 0.044±0.003* |
| T-NOS (U/mg prot) | 19.165±5.252 | 24.377±11.640 | 20.323±11.663 | 23.544±3.765 |
| SOD (U/mg prot) | 7.678±1.058 | 6.193±1.031 | 8.612±2.879 * | 6.037±0.918 |

Values are represented as means ± S.D (n = 3), and the * denotes significant difference ($P<0.05$) comparing with the CN group. CN, the mussel with complete shell and fed in normal sea water (pH 8.1); DN, the mussel with drilled shell and fed in normal sea water (pH 8.1); CA, the mussel with complete shell and fed in acidified sea water (pH 7.4) with exposure time of 48 h; DA, the mussel with drilled shell and fed in acidified sea water (pH 7.4) with exposure time of 48 h.

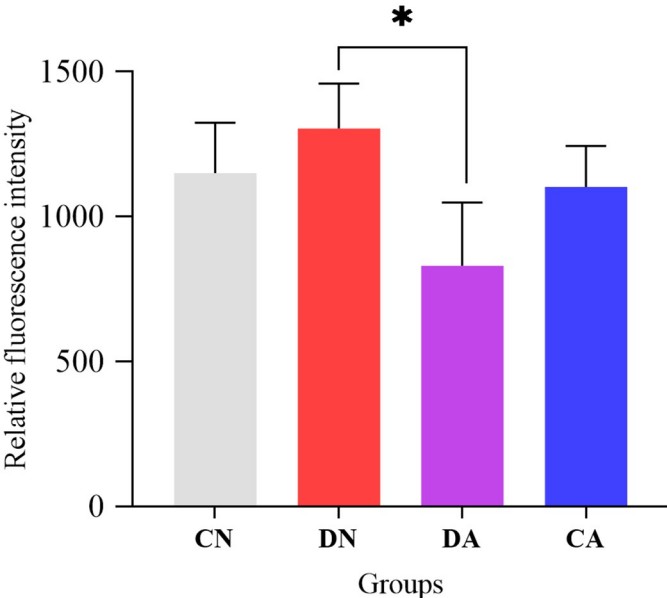

**Fig 7. Intracellular Ca²⁺ level detection in the mantle edge of CN, DN, CA and DA groups.** Vertical bars were represented by mean ± S.D (n = 3), and asterisk denotes significantly difference ($P<0.05$). CN, the mussel with complete shell and fed in normal sea water (pH 8.1); DN, the mussel with drilled shell and fed in normal sea water (pH 8.1); CA, the mussel with complete shell and fed in acidified sea water (pH 7.4) with exposure time of 48 h; DA, the mussel with drilled shell and fed in acidified sea water (pH 7.4) with exposure time of 48 h.

layer with many invaginations can be observed at the surface of the three mantle folds (Figs 8 and 9, and S7 Fig), and our histological observations are similar to mantle tissues from other mollusk, in which, most bivalve mantles have three folds at the edge area [46, 47].

ARS staining further revealed the localization of Ca²⁺ with orange-red color in the columnar epidermal layer of the mantle edge (Fig 8). ARS staining has been used extensively to detect protoplasmic Ca²⁺ (generally orange-red or reddish-yellow) in tissues or cells [48, 49]. In this study, shell damage and/or acute acidification changed the localization of the orange-red area, and the epithelial cell morphology of the outer fold. As shown in Fig 8, unlike the homogeneous distribution of the orange-red colored area in the whole epithelium of the mantle edge from the CN group, the colored area for the mantle edge of the DN, CA, and DA groups showed localization mainly at the top of the epithelium (DN, CA, and DA groups), connective tissue (CA group), or basement area beneath the epithelium layer (DA group) (Fig 8), indicating changes in the secretory behavior of the outer fold epithelium induced by shell damage and/or acute acidification. Furthermore, the cell morphology also changed in the outer fold epithelium, as the intensity of the epidermal cell surface was destroyed in the mussel under shell damage and/or acute acidification (Fig 8). No marked changes in color and epidermal cell morphology was observed in the CN, DN, CA, and DA groups, indicating no noticeable effect of shell damage and/or acute acidification on these two folds (S7 Fig).

Von Kossa staining is used for detecting calcium deposits in tissues, indicated by a black or dark brown coloration in the presence of calcium carbonate or calcium phosphate [50]. In this study, presumptive calcium deposits with dark brown to black coloration were observed on the surface of the epithelial layer of the three mantle folds, and the nuclei and other tissue elements were stained blue to red (Fig 9). Both shell damage and acute acidification induced irregular granular dark brown or black precipitates at the top surface of the epithelial layer,

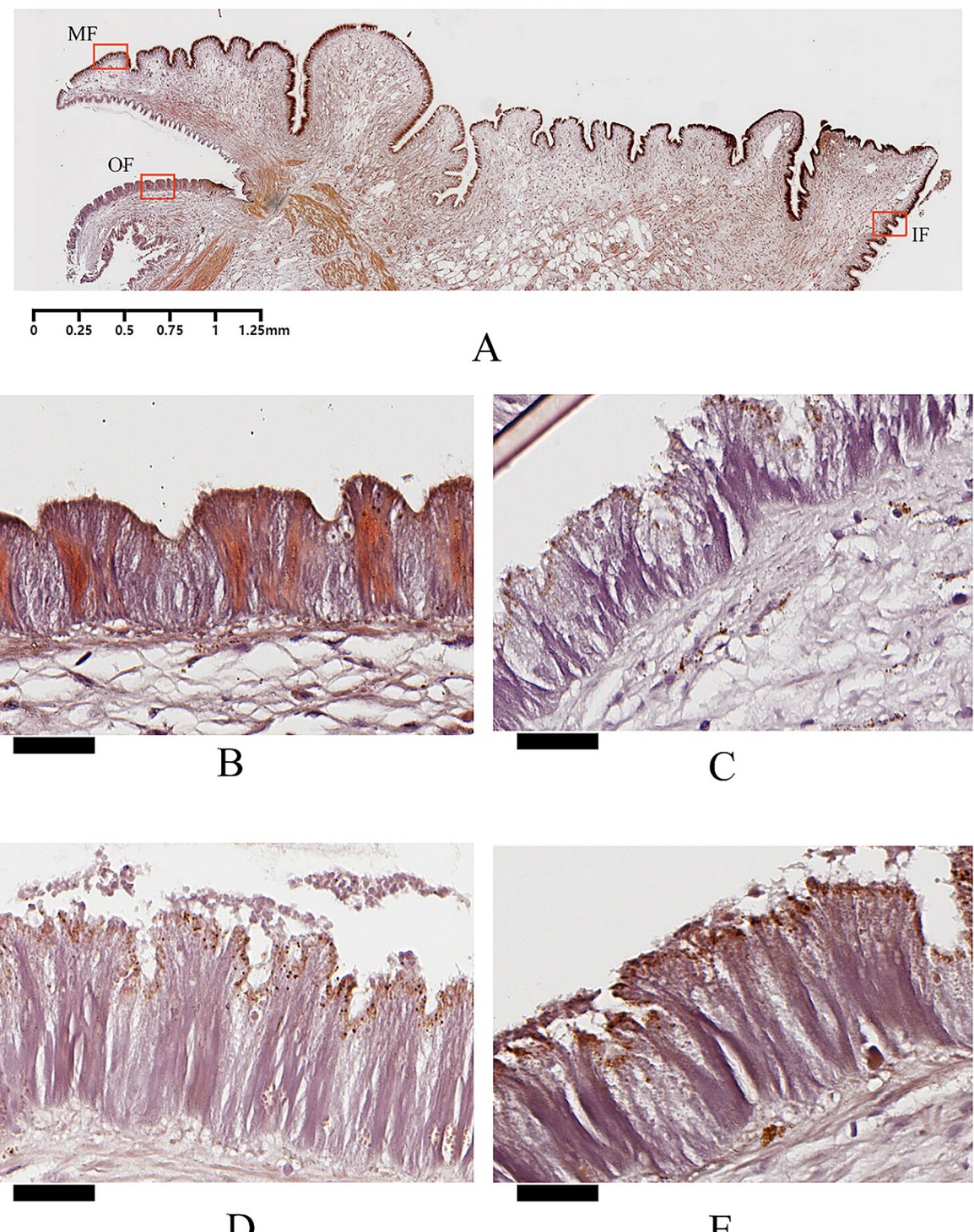

**Fig 8. The relative distribution of intracellular Ca²⁺ in epithelium.** A: the mantle edge was cut at 4μm and stained with Alizarin Red. Three folds (denoted by a red frame) can be observed, including outer fold (OF), middle fold (MF) and inner fold (IF). B: Enlargement of the outer fold from CN group. C: Enlargement of the outer fold from DN group. D: Enlargement of the outer fold from CA group. E: Enlargement of the outer fold from DA group. The scar bar is 1.25 mm for A, and 25 μm for B, C, D, and E.

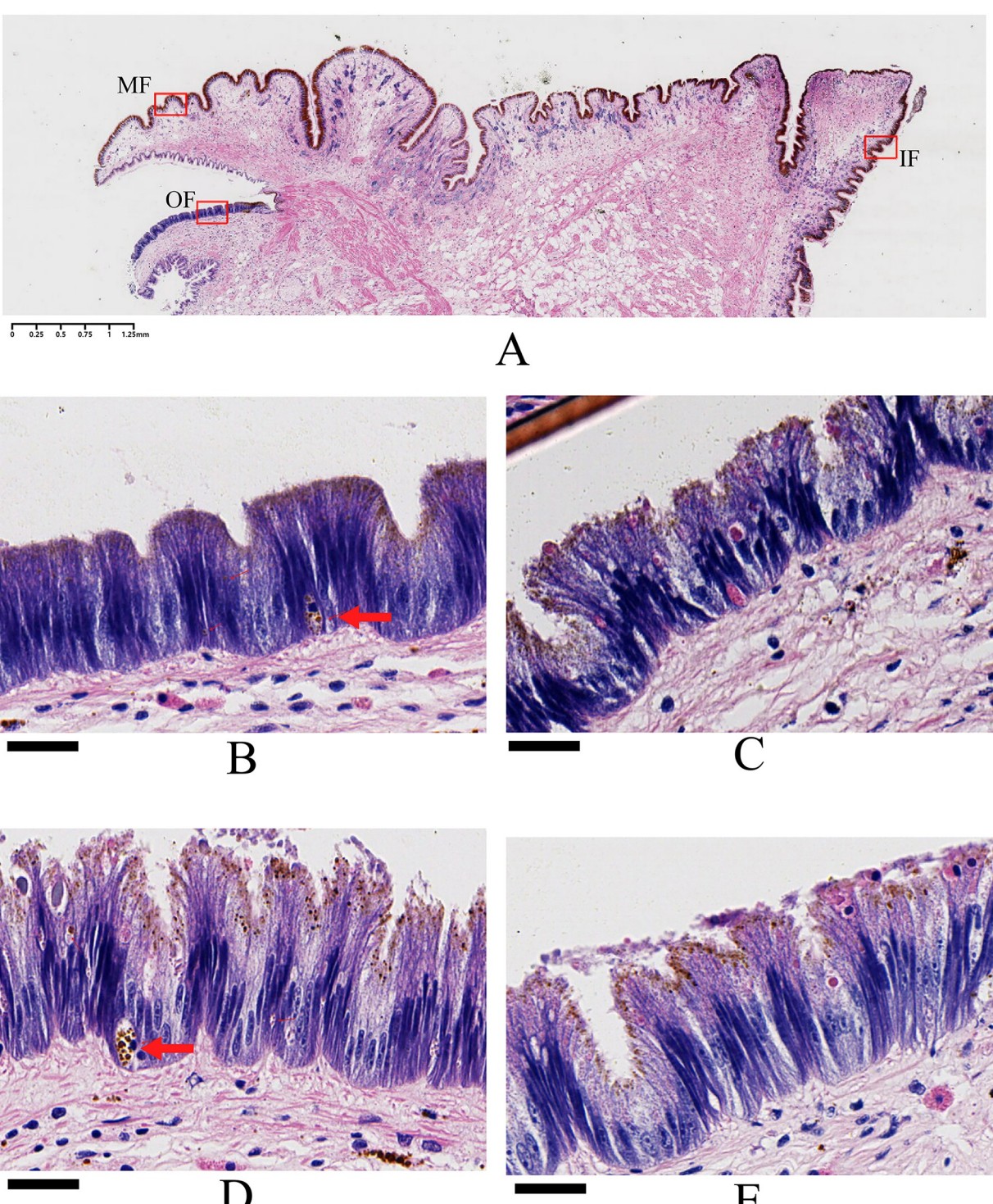

**Fig 9. The relative distribution of insoluble calcium carbonate salts in epithelium.** A: the mantle edge was cut at 4μm and stained with Von Kossa, and the insoluble calcium carbonate salts are dark (brown) colored. Three folds (denoted by a red frame) can be observed, including outer fold (OF), middle fold (MF) and inner fold (IF). B: Enlargement of the outer fold from CN group. C: Enlargement of the outer fold from DN group. D: Enlargement of the outer fold from CA group. E: Enlargement of the outer fold from DA group. The red arrows denote the location of insoluble calcium carbonate salts in epithelium. The scar bar is 1.25 mm for A, and 25 μm for B, C, D, and E.

suggesting that insoluble calcium salt secretion occurred under shell damage and acute acidification (Fig 9). Furthermore, some dark brown granular precipitates surrounded by a membrane were observed inside the bottom of the epithelial cells of the mussels in normal sea water (such as in the CN and DN groups) (Fig 9). Similar to the results of the ARS staining, morphological changes were observed in the epithelium of the mantle outer folds in the DN, CA, and DA groups. The inner and middle folds showed no marked changes in color or morphology, suggesting that shell damage or acute acidification did not affect these two folds (S7 Fig).

## Discussion

Over the last few decades, numerous studies have investigated the impact of OA on marine species and communities, particularly those inhabiting dynamic coastal ecosystems [51]. Because of eutrophication and freshwater inputs, coastal acidification is often much more intense than OA acidification. For example, the coastwide annual mean pH change was estimated at $-0.0085 \pm 0.0069$ unit·yr$^{-1}$ in the past few decades at Hong Kong Coast (Pearl River Estuary), which was over four times stronger than the current estimation of open ocean acidification rate ($\sim$$-0.0019$ unit·yr$^{-1}$) [52]. Similarly, the seawater pH value of the Yangtze River Estuary is rapidly reduced in summer owing to freshwater input [35]. In this study, *M. coruscus* was collected from the Yangtze River Estuary, and it was observed that *M. coruscus* experienced seasonal fluctuations in water pH. Shell fracture or injury frequently occurs for bivalves owing to biotic and abiotic stressors [28, 30]. For *Mytilus*, the shells are continually damaged by the collision and friction between individuals during the high-density farming process, or between the mussels and reef under wave-hurled projectiles [31]. Therefore, *Mytilus* may have evolved shell repair mechanisms to counteract shell injuries. Shell repair processes in bivalve mollusks have been studied in several species [32, 33]. Common mechanisms, including periostracum-like layer formation, calcite secretion, and aragonite formation, have been identified to play a role in the shell repair process in *Mytilus* and other bivalves, and the mantle, especially the mantle edge region, plays a key role in this process [33, 34]. However, the metabolic mechanisms and pathways that respond to shell damage in acidified sea water have not been elucidated. Here, we utilized intact-shell and damaged-shell individuals of *M. coruscus* to investigate the changes in metabolic profiles in the mantle induced by shell damage and/or acute sea water acidification, and how the mantle responds to acute acidification at the beginning of the shell repair process. UPLC-MS/MS analysis revealed that both shell damage and acute acidification disturbed the metabolic processes in the mantle tissue. PCA and OPLS-DA results further revealed significant metabolic changes in mussel mantles under the different treatments (Figs 2 and 3). Based on the number of SDMs among the various comparisons, we observed that in the mantle, more SDMs were induced by acute acidification than by shell damage, indicating a stronger response of the mussel mantle to acute acidification (Fig 4).

The SDMs in the various comparisons were mainly composed of lipids, amino acids, nucleotides, organic acids, benzenoids, and their analogs or derivatives. Shell damage and acute acidification induce SDMs with different patterns. For example, glycylproline, spicamycin, and 2-aminoheptanoic acid (2-AHA) were up-regulated in DN *vs*. CN comparison. Glycylproline belongs to the family of proline-containing peptides [53], and previous studies have confirmed the protective role of this dipeptide during the cellular damage process through wound healing, and cell and tissue regeneration [54–56]. Spicamycin is a nucleoside antibiotic containing fatty acids of various chain lengths (C12-C18), and shows potent antitumor activity against certain human tumor cell lines in vitro [57, 58]. Interestingly, an analog of spicamycin was reported to inhibit osteoclasts and prevent bone destruction in mice with severe immunodeficiency [59]. This suggests that spicamycin may be a potent metabolite in the mantle crucial

for repairing shell damage. 2-aminoheptanoic acid (2-AHA) was identified as an up-regulated metabolite in the DN *vs*. CN comparison. 2-AHA is an aliphatic amino acid molecule with a highly polar "head" and a hydrophobic side chain, making it an amphiphilic molecule. Previous studies have revealed the bilayer-formation ability of 2-AHA [60], and the functions of this molecule in promoting protein synthesis as a Met donor [61] and as an anti-apoptotic agent [62]. These findings suggest possible roles of 2-AHA in shell repair processes, such as periostracum-like layer formation [34], shell matrix protein translation, and regulation of cell apoptosis in mussels under adverse stress conditions.

Some up-regulated metabolites are presented explicitly in response to acute acidification, such as betaine, Asp, and oxidized glutathione (GSSG) in the CA and DA groups. Betaine is an important osmotic regulator with antioxidative or anti-inflammatory activity, and is a methyl group donor [21, 63]; it is considered the most important osmolytes in *Mytilus* for maintaining cell volume under variable salinity or other stresses [64]. The increase in betaine in the mussel mantle indicated osmotic pressure from acute acidification indicating that it may help cope with external osmotic stress. Similar results have been reported in fish [65] and shrimp [66] under different salinity stresses. Asp is a proteinogenic amino acid and an important metabolite of energy metabolism pathways, such as the tricarboxylic acid cycle (TCA). Some Asp-rich shell matrix proteins, such as Asprich, have been reported to be major organic matrices in bivalve shells [67, 68], emphasizing the fundamental importance of Asp in mineralization. The increase in Asp in both the CA and DA groups indicates the rapid adaptation of the mussel to fluctuating pH conditions in estuarine habitats, and more Asp may provide materials for shell matrix protein synthesis. The up-regulation of GSSG under acute acidification indicated that the mantle suffered from oxidative stress caused by acute acidification, and reduced glutathione (GSH) was transformed into oxidized GSSG. Interestingly, we observed that GSH was also up-regulated in the CA and DA groups (Table 2), suggesting the activation of the GSH/GSSG redox under the tested stress levels. Similar results have been observed in other marine invertebrates, such as cephalopods, showing increased reactive oxygen species (ROS) content with enhanced glutathione S-transferase and glutathione reductase activities at low pH [69].

Disturbance of the redox system caused by acute acidification of the mantle was also confirmed by an increase in CAT activity (Table 2). OA induced antioxidative responses of marine invertebrates are species-specific. For example, a low-pH stressor increase SOD and CAT activities in the Yesso Scallop [70], had no effect on CAT and GST activities in soft coral [71], and decreased SOD and CAT activities in *Apostichopus japonicus* [72]. These findings indicated that the physiological redox balance in acidified seawater can be modulated through different mechanisms in different species. In addition, decreased CAT activity in the DN group (Table 2) indicated that different antioxidant mechanisms under shell damage and acute acidification of the mussel mantle and the antioxidative response of the mantle may be suppressed during the shell repair process. T-AOC assay further confirmed that decreased T-AOC was observed in both the DN and DA groups (Table 2).

Both shell damage and acute acidification altered the concentrations of some nucleotides and amino acids (Fig 5 and Table 3), resulting in the enrichment of related KEGG pathways, such as purine metabolism and arginine biosynthesis (Fig 6 and Table 3). Up regulation of GDP, ADP, and AMP and downregulation of adenosine and guanosine were observed in the DN, CA, and DA groups (Table 3). However, the enzymatic activity of 5'-NT presented no significant changes in the DN, CA, and DA groups (Table 2). 5'-NT is an ectoenzyme responsible for the hydrolysis of ATP to adenosine, thus controlling the concentration of nucleotides required for purinergic signal transduction [73]. Therefore, the alteration of nucleotides induced by shell damage and acute acidification may not result from 5'-NT hydrolysis, but is more likely due to the energy consuming process of mussel mantle in response to shell repair

**Table 3. Metabolic pathways enriched by SDMs in six pairwise comparisons of the mantle samples.**

| Metabolic pathways | SDMS (DN vs CN) | SDMS (DA vs CN) | SDMS (CA vs CN) | SDMS (DA vs DN) | SDMS (DA vs CA) |
|---|---|---|---|---|---|
| Purine metabolism | (1.226) Adenosine 3'-monophosphate↑<br>(0.376) Deoxyguanosine) ↓<br>(1.275) dGDP↑<br>(0.621) Guanine↓<br>(0.591) Guanosine↓<br>(0.402) Inosine↓<br>(0.661) Adenosine↓<br>(1.460) ADP↑<br>(1.251) Adenosine monophosphate<br>(0.443) Ribose 1-phosphate↑ | (1.436) Adenosine 3'-monophosphate↑<br>(0.393) Deoxyguanosine↓<br>(1.390) dGDP↑<br>(0.418) Guanine↓<br>(0.384) Guanosine↓<br>(1.259) Guanosine 3'-phosphate↑<br>(0.450) Inosine↓<br>(0.415) Adenosine↓<br>(1.497) ADP↑<br>(1.524) Adenosine monophosphate↑ | (0.625) Adenine↓<br>(1.627) Adenosine 3'-monophosphate↑<br>(0.150) Deoxyadenosine monophosphate↓<br>(0.338) Deoxyguanosine↓<br>(1.460) dGDP↑<br>(0.464) Guanine↓<br>(0.451) Guanosine↓<br>(1.290) Guanosine 3'-phosphate↑<br>(0.519) Inosine↓<br>(1.598) ADP↑<br>(1.651) Adenosine monophosphate↑<br>(0.445) Adenosine↓<br>(2.068) Adenosine 3',5'-diphosphate↑ | (1.171) Adenosine 3'-monophosphate↑<br>(0.673) Guanine↓<br>(0.649) Guanosine↓<br>(1.219) Adenosine monophosphate↑<br>(1.712) Adenosine 3',5'-diphosphate↑<br>(2.505) Ribose 1-phosphate↑ | |
| Glycerophospholipid metabolism | (1.666) LysoPC(20:4 (5Z,8Z,11Z,14Z)/0:0) ↑<br>(2.224) LysoPC(22:6 (4Z,7Z,10Z,13Z,16Z,19Z)) ↑<br>(1.833) LysoPC(20:4 (8Z,11Z,14Z,17Z)) ↑<br>(1.679) LysoPC(20:5 (5Z,8Z,11Z,14Z,17Z)/0:0) ↑<br>(1.292) LysoPC(22:2 (13Z,16Z)/0:0) ↑<br>(2.539) PC (22:5 (4Z,7Z,10Z,13Z,16Z)/0:0) ↑<br>(0.661) Phosphocholine↓<br>(3.511) PE (18:3(6Z,9Z,12Z)/P-18:1(9Z)) ↑<br>(2.581) PE (22:2(13Z,16Z)/20:1(11Z)) ↑<br>(11.828) PC (20:3 (8Z,11Z,14Z)/22:4 (7Z,10Z,13Z,16Z)) ↑<br>(4.987) PC (22:5 (4Z,7Z,10Z,13Z,16Z)/14:1 (9Z)) ↑<br>(2.202) PC (14:0/20:4 (8Z,11Z,14Z,17Z)) ↑<br>(0.556) PC (18:1(11Z)/18:1 (11Z)) ↓<br>(5.429) PC (20:5 (5Z,8Z,11Z,14Z,17Z)/22:4 (7Z,10Z,13Z,16Z)) ↑<br>(0.602) PA (16:0/18:1(11Z)) ↓ | (1.541) LysoPC(18:1 (11Z)/0:0) ↑<br>(1.669) PC (16:1(9Z)/0:0) ↑<br>(1.471) LysoPC(20:2 (11Z,14Z)/0:0) ↑<br>(0.595) Phosphocholine↓<br>(0.643) PC (18:1(11Z)/18:1(11Z)) ↓<br>(2.642) PC (14:0/20:4 (8Z,11Z,14Z,17Z)) ↑<br>(9.422) PC (20:3 (8Z,11Z,14Z)/22:4 (7Z,10Z,13Z,16Z)) ↑<br>(6.152) PC (22:5 (4Z,7Z,10Z,13Z,16Z)/14:1(9Z)) ↑<br>(5.301) PC (20:5 (5Z,8Z,11Z,14Z,17Z)/22:4(7Z,10Z,13Z,16Z)) ↑<br>(3.057) PE (18:3 (6Z,9Z,12Z)/P-18:1 (9Z)) ↑<br>(2.599) PE (22:2 (13Z,16Z)/20:1(11Z)) ↑<br>(0.503) PA (16:0/18:1 (11Z)) ↑ | (2.242) LysoPC (15:0) ↑<br>(1.940) LysoPC(18:1(11Z)/0:0) ↑<br>(1.576) LysoPC (20:4 (5Z,8Z,11Z,14Z)/0:0) ↑<br>(1.764) LysoPC(20:2 (11Z,14Z)/0:0) ↑<br>(1.998) LysoPC 22:6 4Z,7Z,10Z,13Z,16Z,19Z)) ↑<br>(2.194) PC (16:1(9Z)/0:0) ↑<br>(2.938) PE (22:2(13Z,16Z)/20:1(11Z)) ↑<br>(1.367)8-Isoprostaglandin F2a↑<br>(1.629) (9S,10E,12Z)-9-hydroperoxy-10,12-octadecadienoate↑<br>(3.565) Sphingosine↑<br>(1.621) Spermine↑<br>(1.599) L-Arginine↑<br>(4.125) (15Z)-tetracosenoate↑ | | (0.665) LysoPC(20:4 (5Z,8Z,11Z,14Z)/0:0) ↓<br>(0.511) PC (22:5 (4Z,7Z,10Z,13Z,16Z)/0:0) ↓<br>(0.663) LysoPC(20:4 (8Z,11Z,14Z,17Z)) ↓<br>(0.781) PC (20:5 (5Z,8Z,11Z,14Z,17Z)/22:4(7Z,10Z,13Z,16Z)) ↓ |
| Sphingolipid metabolism | (1.383) Cer(d18:1/16:0) ↑<br>(0.797) SM (d18:0/14:1(9Z) (OH)) ↓ | (2.003) Sphinganine↑<br>(4.967) Sphingosine↑<br>(1.367) Cer (d18:1/16:0) ↑ | (2.381) Sphinganine↑<br>(3.565) Sphingosine↑ | (0.410) 3-Dehydrosphinganine↓<br>(1.872) Sphinganine↑<br>(5.123) Sphingosine↑ | |
| Cysteine and methionine metabolism | (0.748) Methionine sulfoxide↓<br>(0.789) Phosphohydroxypyruvic acid↓<br>(0.571) L-Homocysteine↓ | | | | |

*(Continued)*

**Table 3.** (Continued)

| Metabolic pathways | SDMS (DN vs CN) | SDMS (DA vs CN) | SDMS (CA vs CN) | SDMS (DA vs DN) | SDMS (DA vs CA) |
|---|---|---|---|---|---|
| Linoleic acid metabolism | (11.828) PC (20:3 (8Z,11Z,14Z)/22:4 (7Z,10Z,13Z,16Z)) ↑ (4.987) PC (22:5 (4Z,7Z,10Z,13Z,16Z)/14:1 (9Z)) ↑ (2.202) PC (14:0/20:4 (8Z,11Z,14Z,17Z)) ↑ (0.556) PC (18:1(11Z)/18:1 (11Z)) ↓ (5.429) PC (20:5 (5Z,8Z,11Z,14Z,17Z)/22:4 (7Z,10Z,13Z,16Z)) ↑ (0.602) PA (16:0/18:1(11Z)) ↓ (1.170) (9S,10E,12Z)-9-hydroperoxy-10,12-octadecadienoate↑ | | | (1.236) (9S,10E,12Z)-9-hydroperoxy-10,12-octadecadienoate↑ (0.743) bishomo-gamma-linolenic acid↓ (1.236) (9S,10E,12Z)-9-hydroperoxy-10,12-octadecadienoate↑ (0.050) PC (22:0/20:0) ↓ (0.871) PC (22:5 (4Z,7Z,10Z,13Z,16Z)/16:1 (9Z)) ↓ (0.797) PC (20:3 (8Z,11Z,14Z)/22:4 (7Z,10Z,13Z,16Z)) ↓ | |
| Aminoacyl-tRNA biosynthesis | | (0.582) L-Alanine↓ (1.334) L-Lysine↑ (1.876) L-Aspartic Acid↑ (1.341) L-Arginine↑ (1.446) L-Tyrosine↑ (1.335) L-Isoleucine↑ | (1.404) L-Lysine↑ (1.423) L-Proline↑ (1.599) L-Arginine↑ (1.978) L-Aspartic Acid↑ (1.570) L-Tyrosine↑ (1.339) L-Isoleucine↑ | (0.702) L-Alanine↓ (1.393) L-Lysine↑ (1.745) L-Aspartic Acid↑ (1.356) L-Proline↑ (1.417) L-Tyrosine↑ (1.346) L-Isoleucine↑ | |
| ABC transporters | | (0.660) N-Acetyl-D-glucosamine↓ (1.177) Betaine↑ (0.582) L-Alanine↓ (1.334) L-Lysine↑ (1.876) L-Aspartic Acid↑ (1.341) L-Arginine↑ (0.415) Adenosine↓ | (0.445) Adenosine↓ (1.599) L-Arginine↑ (1.404) L-Lysine↑ (1.978) L-Aspartic Acid↑ (1.240) Betaine↑ (0.260) 4-Hydroxyproline↓ | (0.740) N-Acetyl-D-glucosamine↓ (0.317) 4-Hydroxyproline↓ (1.137) Betaine↑ (0.628) Adenosine↓ (0.702) L-Alanine↓ (1.393) L-Lysine↑ (1.745) L-Aspartic Acid↑ | |
| Neuroactive ligand-receptor interaction | | (0.403) Palmitoyl Ethanolamide↓ (2.815) Leukotriene C4↑ (1.876) L-Aspartic Acid↑ (0.415) Adenosine↓ | (0.445) Adenosine↓ (1.978) L-Aspartic Acid↑ (0.360) Palmitoyl Ethanolamide↓ (3.181) Leukotriene C4↑ | (0.384) Palmitoyl Ethanolamide↓ (1.812) Leukotriene C4↑ (0.628) Adenosine↓ (1.745) L-Aspartic Acid↑ | |
| Alanine, aspartate and glutamate metabolism | | (2.237) Citric acid↑ (0.582) L-Alanine↓ (1.876) L-Aspartic Acid↑ | | (0.702) L-Alanine↓ (1.745) L-Aspartic Acid↑ (2.004) Citric acid↑ | |

(*Continued*)

**Table 3.** (*Continued*)

| Metabolic pathways | SDMS (DN vs CN) | SDMS (DA vs CN) | SDMS (CA vs CN) | SDMS (DA vs DN) | SDMS (DA vs CA) |
|---|---|---|---|---|---|
| Arachidonic acid metabolism | | (1.264)14,15-Epoxy-5,8,11-eicosatrienoic acid↑ (0.796)15-F2t-IsoP↓ (3.410)20-Hydroxy-leukotriene E4↑ (0.643) PC (18:1(11Z)/18:1(11Z)) ↓ (2.642) PC (14:0/20:4 (8Z,11Z,14Z,17Z)) ↑ (9.422) PC (20:3 (8Z,11Z,14Z)/22:4 (7Z,10Z,13Z,16Z)) ↑ (6.152) PC (22:5 (4Z,7Z,10Z,13Z,16Z)/14:1(9Z) ↑) (5.301) PC (20:5 (5Z,8Z,11Z,14Z,17Z)/22:4(7Z,10Z,13Z,16Z)) ↑ | (0.697) 15-F2t-IsoP↓ (1.367)8-Isoprostaglandin F2a↑ (4.156)20-Hydroxy-leukotriene E4↑ (4.578) LTE4↑ (10.762) PC (20:3 (8Z,11Z,14Z)/22:4 (7Z,10Z,13Z,16Z)) ↑ (2.980) PC (14:0/20:4 (8Z,11Z,14Z,17Z)) ↑ (6.786) PC (20:5 (5Z,8Z,11Z,14Z,17Z)/22:4 (7Z,10Z,13Z,16Z)) ↑ (6.850) PC (22:5 (4Z,7Z,10Z,13Z,16Z)/14:1 (9Z)) ↑ (0.610) PC (18:1(11Z)/18:1 (11Z)) ↓ | | |
| Pantothenate and CoA biosynthesis | | | (1.621) Spermine↑ (1.978) L-Aspartic Acid↑ (2.068) Adenosine 3',5'-diphosphate↑ | (1.534) Spermine↑ (1.745) L-Aspartic Acid↑ (1.712) Adenosine 3',5'-diphosphate↑ | |
| Arginine biosynthesis | | | (1.432) N-alpha-Acetyl-L-citrulline↑ (1.978) L-Aspartic Acid↑ (1.599) L-Arginine↑ | | |
| Citrate cycle | | | | (2.004) Citric acid↑ (1.363) Isocitrate↑ (1.242) Malic acid↑ | |
| Biosynthesis of unsaturated fatty acids | | | | (2.332) (15Z)-tetracosenoate↑ (1.500) Behenic acid↑ (0.860) Docosadienoic acid↓ (0.723) Eicosapentaenoic acid↓ (0.743) bishomo-gamma-linolenic acid↓ | |
| Pyruvate metabolism | | | | | (1.363) L-Lactic acid↑ (0.765) Malic acid↓ |
| Inositol phosphate metabolism | (0.552) D-Myoinositol-4-phosphate ↓ | (2.150) D-Myoinositol-4-phosphate ↑ | (2.008) D-Myoinositol-4-phosphate ↑ | (1.546) D-Myoinositol-4-phosphate ↑ | |

CN, the mussel with complete shell and fed in normal sea water (pH 8.1); DN, the mussel with drilled shell and fed in normal sea water (pH 8.1); CA, the mussel with complete shell and fed in acidified sea water (pH 7.4) with exposure time of 48 h; DA, the mussel with drilled shell and fed in acidified sea water (pH 7.4) with exposure time of 48 h

and acute acidification and more ATP is required to fill the energy gap. The impact of OA on the energy metabolism of bivalves had been reported previously, and massive ATP consumption in the mantle of bivalves was observed, indicating that a large amount of energy was allocated to maintain the acid-base balance in a reduced-pH environment [70, 74]. In addition, both shell damage and acute acidification decrease the content of propionylcarnitine (PLC), a molecule involved in both carbohydrate and lipid metabolism [75]. Considering the positive roles of propionate in the TCA cycle and carnitine in lipid transportation and beta-oxidation,

the lysis of propionylcarnitine has been reported to be involved in energy generation [76]. Therefore, the decrease in PLC may result from the lysis of this molecule to provide more propionate and carnitine for rapid energy regulation in the mussel mantle undergoing shell damage and/or acute acidification and an increase in phospholipids and tricarboxylic acid cycle related metabolites, such as malic acid (malate) is observed (Fig 5). These findings highlight the energy consumption of mussels in response to environmental stress. Similar findings have been reported in other bivalves [36, 37].

Amino acid metabolism is significantly altered by shell damage and/or acute acidification. Upregulated Arg, Gly, Glu, and Ile were observed in the comparisons of DN *vs*. CN and CA *vs*. CN (Fig 5 and Table 1), and the corresponding KEGG pathways in amino acid metabolism were enriched in these comparisons, including alanine, aspartate, and glutamate metabolism, lysine biosynthesis, and D-arginine and D-ornithine metabolism (Fig 6 and Table 3). FAA analysis further confirmed the upregulation of Arg, Glu, and Gly in response to shell damage and/or acute acidification (Table 1). Increases in intracellular free amino acid levels occur mainly through amino acid synthesis or protein degradation pathways. Similar findings have been reported previously; for example, up-regulation of alanine and aspartate in *Crassostrea gigas* under elevated pCO2 exposure [77], upregulation of tyrosine, and down-regulation of glycine in *M. coruscus* hemolymph under 7–21 d exposure to low pH [78]. Arginine and its metabolic pathways are important modulators of several physiological processes in animals, including immune responses [79], signal transduction [80], and ammonia excretion *via* urea and polyamine formation [81]. In this study, T-NOS detection revealed that the enzymatic activity showed no significant change under shell damage and/or acute acidification (Table 2), suggesting that an increase in Arg did not induce NO synthesis. Notably, urea levels were upregulated in the CA group (Table 1). Arg is the precursor of urea, and the upregulation of both Arg and urea indicates the possible activation of the urea cycle in the mussel mantle under shell damage or acute acidification. Urea/urease-aided $CaCO_3$ mineralization had been reported previously. The process takes advantage of the supply of $CO_3^{2-}$ ions derived from urea hydrolysis and an increase in the pH generated by the reaction, the presence of $Ca^{2+}$ ions leads to the precipitation of $CaCO_3$ [82]. The elevation of both Arg and urea implied that the mussel mantle may use urea to generate $CO_3^{2-}$ and $NH_4^+$ ions, thereby promoting shell repair and increasing the local pH value. However, this hypothesis warrants further investigation. Among the other upregulated amino acids, Glu is a ubiquitous amino acid related to formation of alpha-ketoglutarate, an integral component of the TCA cycle [83], and plays a vital role in the energy supply. Glu plays an important role in $Ca^{2+}$ transportation [84, 85]. Gly and its products, glycine and betaine, are involved in the formation and regulation of osmotic pressure in marine animals [66]. The upregulation of Glu and Gly indicates potential positive responses of the mussel mantle to both shell damage and acute acidification. Upregulated Glu may increase the TCA cycle and $Ca^{2+}$ transport during shell repair, and Gly may provide more material for osmoregulation under acute acidification.

The reduction in $CaCO_3$ saturation in sea water caused by OA can adversely affect the calcification and growth of marine bivalves [86–88]. Amorphous calcium carbonate (ACC), a precursor for shell biomineralization in bivalves, is probably deposited in the mantle and contributes to the repair and formation of shells in bivalves [89, 90]. Previous studies have demonstrated that OA induces ACC formation in mussels, suggesting that ACC can be used as a repair mechanism to combat shell damage caused by OA [18]. In this study, both ARS and von Kossa staining revealed that the secretion of calcium was significantly altered in the epithelial layer of the mantle outer fold, as more stained calcium deposits were observed on the surface of the outer fold epithelium in the DN, CA, and DA groups (Figs 8 and 9). Although the crystal form of these calcium particles could not be specifically determined in this study, our

histological observations highlighted that the outer fold is the main site responsible for the mussel shell repair process, and ACC-like calcium particles may be secreted from the outer fold, thus provide more calcium carbonate precursors for shell repair under acute acidification. Similar findings have been previously reported for other marine calcifies, such as mussels and corals [18, 91, 92]. Furthermore, D-Myoinositol-4-phosphate, which is a cyclitol inositol with important roles in intracellular signal transduction [93], membrane construction, and trafficking in all eukaryotes [94], was significantly upregulated in the mantle of the CA and DA groups (Table 3). Cell morphological changes, such as membrane damage and invagination at the epithelium of the outer fold in DN, CA, and DA groups (Figs 8 and 9) may be associated with the up-regulation of D-Myoinositol-4-phosphate in these groups. In addition, D-Myoinositol-4-phosphate is also a downstream metabolite of inositol-1,4,5-trisphosphate (IP3) [95, 96], and IP3 is hydrolyzed from phosphatidylinositol-4,5-bisphosphate (PIP$_2$), thus forming a classical pathway for regulating cellular $Ca^{2+}$ concentrations [97]. In bivalves, mantle cells help sequester and concentrate $Ca^{2+}$ ions into the endoplasmic reticulum to maintain cytosolic $Ca^{2+}$ homeostasis [90, 98]. The $Ca^{2+}$ ions can be released from the cell endoplasmic reticulum into the cytosol and are further released into the extracellular matrix *via* the classical PIP$_2$/IP3 pathway and its activated $Ca^{2+}$-ATP enzyme if necessary [99]. Intracellular $Ca^{2+}$ levels were measured using a Fluo-3 fluorescence probe and a flow cytometer. We noted that intracellular $Ca^{2+}$ levels of the mantle cells showed a slight increasing under shell damage (Fig 7), indicating that shell damage may induce more $Ca^{2+}$ ions to be recruited in mantle cells *via* an unclear mechanism. Acute acidification showed no change in the intracellular $Ca^{2+}$ level compared to that of the control group (CN group) (Fig 7), indicating an intracellular homeostasis of $Ca^{2+}$ level for the mussel mantle cells even under low environmental pH. This is important for the mussels that face elevated $Ca^{2+}$ levels due to acute acidification in sea water when the balance between intracellular and extracellular $Ca^{2+}$ levels is disrupted. Interestingly, Wang *et al.* [100] recently reported calmodulin-like protein mediated transportation of $Ca^{2+}$ in oysters from hemocytes to mantle cells, and the downregulation of calmodulin expression and the relocation of this molecule into the mantle epithelium resulted in calcium deposition in the mantle epithelium in acidified sea water. The calmodulin protein is involved in IP3-binding and regulating calcium transportation [97]. In addition, calmodulin can inhibit both IP3 and IP3-evoked $Ca^{2+}$ release [101]. According to the findings mentioned above, we speculate that downregulation of calmodulin may activate the IP3 pathway and induce more $Ca^{2+}$ release from the cytosol to the extracellular matrix, enhancing the IP3 pathway, thus increasing D-Myoinositol-4-phosphate level, which was observed in this study. In contrast, low environmental pH induces more calcium depositions in mantle epithelial cells, as previously reported for oysters [100]. We observed calcium deposits on the surface of the mantle outer fold epithelium under acute acidification revealed by ARS staining in this study (Fig 8), confirming a flexible response of *M. coruscus* mantle to acute acidification.

Shell damage and/or acute acidification also alter the lipid metabolism in the mussel mantle. In this study, long-chain phospholipids, such as phosphatidylcholine (PC), phosphatidyl ethanolamine (PE), and pentaenoic acid (PA), were observed up-regulated under both shell damage and acute acidification (Fig 5). PA is a precursor of PC and PE, both of which are involved in regulating the composition and structure of cell membranes [102, 103]. The significant upregulation of these phospholipids in the mussel mantle implies that membrane repair processes continue to function in acidified environments or during shell repair. Considering the cell membrane damage in the mantle outer fold epithelium revealed by ARS and von Kossa staining (Figs 8 and 9), the upregulation of these lipids may be involved in repair of the cell membrane of the mussel mantle under environmental stress. Moreover, lipid related KEGG pathways were also enriched with different patterns in the altered phospholipids in different

comparisons, such as enriched linoleic lipid-metabolism in DN *vs*. CN and sphingolipid metabolism in CA *vs*. CN (Fig 6), indicating different responses of the mussel mantle to shell damage and acute acidification. However, the mechanisms underlying the altered lipid-related metabolism of mussel mantle under different stressors remain to be elucidated.

Some KEGG pathways, such as the mTOR and FoxO signaling pathways, were enriched in SDMs from both shell damage and acute acidification (Fig 6A and 6B), indicating cell functional changes under the stresses tested in the present study. The mTOR signaling pathway is a highly conserved signaling network that regulates cell growth in response to nutrients, hormones, and stresses [104]. mTOR has been reported to regulate autophagy, and activation of the mTOR signaling pathway can inhibit autophagy [105]. We observed that the autophagy pathway was also enriched in SDMs in the DN *vs*. CN (Fig 6), suggesting that the regulation of autophagy may be mTOR-independent in the mussel mantle. However, this hypothesis must be verified in future studies. The FoxO signaling pathway is considered to have inhibitory effect on cell proliferation and has been suggested to play pivotal roles in cell metabolism, growth, differentiation, oxidative stress, senescence, autophagy, and aging [106, 107]. The enriched FoxO signaling pathway induced by both shell damage and acute acidification indicates mantle cell functional changes in mussels in response to the tested stresses.

## Conclusion

In the present study, we used intact-shell and damaged-shell *M. coruscus*, and performed metabolomic analyses of the mantle under normal and acute acidification scenarios to ascertain the molecular responses of the mussel mantle to different stressors. Here, we report that both shell damage and acute acidification induce alterations in phospholipids, amino acids, nucleotides, organic acids, benzenoids, and their analogs or derivatives. Glycylproline, spicamycin, and 2-AHA are specifically induced by shell damage, whereas betaine, Asp, and GSSG are specifically induced by acute acidification. Our results showed different patterns of SDMs in the mussel mantle under different stresses (shell damage, acute acidification, and a combination of these two), thus providing clues for understanding the mussel lunch shell repair process under OA. Furthermore, metabolic process related to energy supply, cell function, signal transduction, and amino acid synthesis are disturbed by shell damage and/or acute acidification. We speculated that both shell damage and acute acidification activated energy consumption, phospholipid synthesis, osmotic regulation, and redox balance. Moreover, FAA analysis and enzymatic activity assays partially confirmed this hypothesis. Our findings highlight the adaptation of *M. coruscus* in estuarine areas with dramatic fluctuations in pH, and may prove instrumental in its ability to continue shell biomineralization and survive OA.

## Supporting information

**S1 Fig. Ion chromatograms of QC samples.** A ~ D: Ion chromatograms of QC samples under positive model for QC1 ~QC4, respectively; E ~ H: Ion chromatograms of QC samples under negative model for QC1 ~QC4, respectively.
(TIF)

**S2 Fig. Metabolite intensity distribution plots for four mantle sample groups.** CN, the mussel with complete shell and fed in normal sea water (pH 8.1); DN, the mussel with drilled shell and fed in normal sea water (pH 8.1); CA, the mussel with complete shell and fed in acidified sea water (pH 7.4) with exposure time of 48 h; DA, the mussel with drilled shell and fed in acidified sea water (pH 7.4) with exposure time of 48 h.
(TIF)

**S3 Fig. Response permutation testing plots from five pairwise comparisons among the mantle samples.** CN, the mussel with complete shell and fed in normal sea water (pH 8.1); DN, the mussel with drilled shell and fed in normal sea water (pH 8.1); CA, the mussel with complete shell and fed in acidified sea water (pH 7.4) with exposure time of 48 h; DA, the mussel with drilled shell and fed in acidified sea water (pH 7.4) with exposure time of 48 h. (A)DN vs CN; (B)DA vs CN; (C)CA vs CN; (D)DA vs DN; (E)CA vs DA.
(TIF)

**S4 Fig. Volcano maps of the metabolites from five pairwise comparisons among the mantle samples.** CN, the mussel with complete shell and fed in normal sea water (pH 8.1); DN, the mussel with drilled shell and fed in normal sea water (pH 8.1); CA, the mussel with complete shell and fed in acidified sea water (pH 7.4) with exposure time of 48 h; DA, the mussel with drilled shell and fed in acidified sea water (pH 7.4) with exposure time of 48 h. (A)DN vs CN; (B)DA vs CN; (C)CA vs CN; (D)DA vs DN; (E)CA vs DA.
(TIF)

**S5 Fig. The standard curve of FAA analysis.**
(TIF)

**S6 Fig. Histogram of fluorescence intensity per 10 000 cells of the mantle samples.** CN, the mussel with complete shell and fed in normal sea water (pH 8.1); DN, the mussel with drilled shell and fed in normal sea water (pH 8.1); CA, the mussel with complete shell and fed in acidified sea water (pH 7.4) with exposure time of 48 h; DA, the mussel with drilled shell and fed in acidified sea water (pH 7.4) with exposure time of 48 h.
(TIF)

**S7 Fig. Histological observation of middle fold (MF) and inner fold (IF) from the mantle edge samples from CN, DN, CA and DA groups.** A: the mantle edge was cut at 4μm and stained with ARS. B: the mantle edge was cut at 4μm and stained with Von Kossa. CN, the mussel with complete shell and fed in normal sea water (pH 8.1); DN, the mussel with drilled shell and fed in normal sea water (pH 8.1); CA, the mussel with complete shell and fed in acidified sea water (pH 7.4) with exposure time of 48 h; DA, the mussel with drilled shell and fed in acidified sea water (pH 7.4) with exposure time of 48 h.
(TIF)

**S1 Table. The model parameter of OPLS-DA plots five pairwise comparisons among the mantle samples.** CN, the mussel with complete shell and fed in normal sea water (pH 8.1); DN, the mussel with drilled shell and fed in normal sea water (pH 8.1); CA, the mussel with complete shell and fed in acidified sea water (pH 7.4) with exposure time of 48 h; DA, the mussel with drilled shell and fed in acidified sea water (pH 7.4) with exposure time of 48 h.
(DOCX)

## Author Contributions

**Investigation:** Xiaojun Fan, Changsheng Tang.

**Methodology:** Ying Wang, Xiaolin Zhang, Jianyu He.

**Project administration:** Xiaojun Yan, Zhi Liao.

**Writing – original draft:** Xiaojun Fan.

**Writing – review & editing:** Isabella Buttino, Xiaojun Yan, Zhi Liao.

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
