## [Decision Letter · Decision Letter 0]

7 Sep 2023

PONE-D-23-22159Metabolic profiling of Mytilus coruscus mantle in response of shell repairing under acute acidificationPLOS ONE

Dear Dr. Liao,

Thank you for submitting your manuscript to PLOS ONE. After careful consideration, we feel that it has merit but does not fully meet PLOS ONE’s publication criteria as it currently stands. Therefore, we invite you to submit a revised version of the manuscript that addresses the points raised during the review process.

Please note that we have only been able to secure a single reviewer to assess your manuscript. We are issuing a decision on your manuscript at this point to prevent further delays in the evaluation of your manuscript. Please be aware that the editor who handles your revised manuscript might find it necessary to invite additional reviewers to assess this work once the revised manuscript is submitted. However, we will aim to proceed on the basis of this single review if possible. The reviewer has raised a number of concerns for your attention, particularly regarding details of the methodology, additional data and clarity regarding comparisons performed as part of this study. Please ensure that you address each of the reviewer's comments when revising your manuscript. We note that the reviewer has recommended that you cite specific previously published works. As always, we recommend that you please review and evaluate the requested works to determine whether they are relevant and should be cited. It is not a requirement to cite these works. We appreciate your attention to this request.

We look forward to receiving your revised manuscript.

Kind regards,

Hugh Cowley

Staff Editor

PLOS ONE

“This study was supported by the National Natural Science Foundation of China (Grant Nos. 32271580, 42020104009, and 32200083). Zhejiang Provincial Natural Science Foundation (Grant Nos. LQ23D060002 and LTGS23C010001).”

“ This research was supported by the National Natural Science Foundation of China (Grant Nos. 32271580, 42020104009, and 32200083). Zhejiang Provincial Natural Science Foundation (Grant Nos. LQ23D060002 and LTGS23C010001).”

“This study was supported by the National Natural Science Foundation of China (Grant Nos. 32271580, 42020104009, and 32200083). Zhejiang Provincial Natural Science Foundation (Grant Nos. LQ23D060002 and LTGS23C010001).”

6. We are unable to open your Supporting Information file [Supporting information.rar]. Please kindly revise as necessary and re-upload.

Reviewers' comments:

Reviewer's Responses to Questions

**Comments to the Author**

1. Is the manuscript technically sound, and do the data support the conclusions?

Reviewer #1: Yes

2. Has the statistical analysis been performed appropriately and rigorously? 

Reviewer #1: Yes

3. Have the authors made all data underlying the findings in their manuscript fully available?

Reviewer #1: Yes

4. Is the manuscript presented in an intelligible fashion and written in standard English?

Reviewer #1: Yes

5. Review Comments to the Author

Reviewer #1: The manuscript entitled “Metabolic profiling of Mytilus coruscus mantle in response of shell repairing under acute acidification” presents reasonable experimental design and data analysis process, and reports results of the physiological impact of shell damage and ocean acidification conditions on Mytilus coruscus. Overall, the authors' findings provide some interesting data for explaining the possible molecular responses of the mussel mantle to the low pH and shell damage, and thusly make it worthy of publication. Some specific comments were provided below. The authors are welcome to follow my suggestions for improving the manuscript.

Introduction:

Introduction is generally elucidative. The cited references are exhaustive and updated, and provide an adequate background regarding the topic of the research.

In a picky way, the following recently published article or book might be appropriate citations to further strengthen the background of this study.

https://doi.org/10.1016/j.aquatox.2020.105740

https://doi.org/10.1016/C2019-0-04274-4

Materials and Methods:

The experimental design needs to be more detailed.

- How was the experimental system calibrated to ensure that the reported pH values (8.1 and 7.4) were precise?

- Additionally, could the authors please provide some type of environmental/ecological justification for the low pH value used?

- The authors stated that there are four groups of mussels used in this study. Was there pH treatment-level replication?

- were drilled/undrilled mussels that were reared in low pH all in the same low pH tank? - How many individuals for each group, and in each tank?

Results:

- The standard curve of FAA analysis used for qualitative and quantitative analysis of free amino acids and other nitrogenous compounds should be provided as supplementary file.

- Also in the section of FAA analysis, the authors used “μg/g wet tissue” as the unit of free amino acids content in Table 1, and “μg/g dry tissue” in other place. Which one is correct?

Discussion:

- The authors should clearly specify which comparisons were performed and base the interpretation of results on those specific comparisons. This will help avoid any misinterpretation of the findings. For example, in Line 415, “The SDMs in various comparisons…”, and Line 445, “The up-regulation of GSSG under the acute acidification…”.

In addition, the following articles might be decent for authors to further strengthen their discussion.

https://doi.org/10.1021/acs.est.1c06735

https://doi.org/10.1016/j.scitotenv.2022.156442

https://doi.org/10.1016/j.chemosphere.2019.125415

Other minor errors:

Some typo errors should be corrected. Such as:

- In the section of Enzymatic assay, “described in the section 2.1”, where is the section 2.1?

- In Line 386, “no mark changes” should be “no marked changes”;

- In Discussion, paragraph six, “overserved” should be “observed”; in the next paragraph, “as one of negative effect for…” should be “as one of negative effects on…”, etc.

6. PLOS authors have the option to publish the peer review history of their article (what does this mean?). If published, this will include your full peer review and any attached files.

Reviewer #1: No

---

## [Author Response · Author response to Decision Letter 0]

28 Sep 2023

We express our gratitude to the reviewers and editors for their valuable comments. We are glad that the reviewers give us the opportunity to upload a revised version of our manuscripts. Below there are our responds to the reviewer´s comments. 

Review Comments to the Author

Reviewer #1: The manuscript entitled “Metabolic profiling of Mytilus coruscus mantle in response of shell repairing under acute acidification” presents reasonable experimental design and data analysis process, and reports results of the physiological impact of shell damage and ocean acidification conditions on Mytilus coruscus. Overall, the authors' findings provide some interesting data for explaining the possible molecular responses of the mussel mantle to the low pH and shell damage, and thusly make it worthy of publication. Some specific comments were provided below. The authors are welcome to follow my suggestions for improving the manuscript.

Introduction:

Introduction is generally elucidative. The cited references are exhaustive and updated, and provide an adequate background regarding the topic of the research.

In a picky way, the following recently published article or book might be appropriate citations to further strengthen the background of this study.

https://doi.org/10.1016/j.aquatox.2020.105740

https://doi.org/10.1016/C2019-0-04274-4

Response: thanks for reviewer’s comments. We checked the references and one of them was added in Introduction as reference 21. 

Materials and Methods:

The experimental design needs to be more detailed.

- How was the experimental system calibrated to ensure that the reported pH values (8.1 and 7.4) were precise?

- Additionally, could the authors please provide some type of environmental/ecological justification for the low pH value used?

- The authors stated that there are four groups of mussels used in this study. Was there pH treatment-level replication?

- were drilled/undrilled mussels that were reared in low pH all in the same low pH tank? - How many individuals for each group, and in each tank?

Response: thanks for reviewer’s suggestions. We rewrote the section and the detail information was added accordingly. 

- In this study, the pH value was monitored using a pH meter, and adjusted using a CO2 pump to ensure the precise of pH value. 

- The two pH levels used in this study were selected based on the average pH value at the local mussel farm and the estuarine habitat where M. coruscus lives in a highly ﬂuctuating pH environments due to river run oﬀ [doi: 10.1016/j.scitotenv.2020.142838.]. The low pH 7.2-7.6 had been observed previously in this area [doi: 10.1007/s11356-022-21122-z.], and pH 7.4 representing coastal acidification had been used for other estuarine species, such as oyster, in previous studies [doi: 10.1111/mec.16751.].

- Furthermore, three pH treatment-level replications were used for the mussels in this study. For each pH treatment-level replication, the mussels with intact-shell and damaged-shell were mixed and raised in the same tank. Six tanks were prepared for the mussels and 30 mussel individuals, including 15 intact-shell and 15 damaged-shell mussels were raised in each tank. A total of 180 mussels were used for our experiment, and 45 individuals for each group. 

Results:

- The standard curve of FAA analysis used for qualitative and quantitative analysis of free amino acids and other nitrogenous compounds should be provided as supplementary file.

- Also in the section of FAA analysis, the authors used “μg/g wet tissue” as the unit of free amino acids content in Table 1, and “μg/g dry tissue” in other place. Which one is correct?

Response: thanks for reviewer’s suggestions.

- The standard curve of FAA analysis was added in the revised manuscript as S5 Fig according to the reviewer’s suggestion.

- We used μg/g of dry tissue as unit of the FAA analysis result, and the “μg/g wet tissue” is a mistake and we corrected accordingly. 

Discussion:

- The authors should clearly specify which comparisons were performed and base the interpretation of results on those specific comparisons. This will help avoid any misinterpretation of the findings. For example, in Line 415, “The SDMs in various comparisons…”, and Line 445, “The up-regulation of GSSG under the acute acidification…”.

In addition, the following articles might be decent for authors to further strengthen their discussion.

https://doi.org/10.1021/acs.est.1c06735

https://doi.org/10.1016/j.scitotenv.2022.156442

https://doi.org/10.1016/j.chemosphere.2019.125415

Response: thanks for reviewer’s suggestions. We revised some ambiguous descriptions in Discussion, and one of the suggested references was added. 

Other minor errors:

Some typo errors should be corrected. Such as:

- In the section of Enzymatic assay, “described in the section 2.1”, where is the section 2.1?

- In Line 386, “no mark changes” should be “no marked changes”;

- In Discussion, paragraph six, “overserved” should be “observed”; in the next paragraph, “as one of negative effect for…” should be “as one of negative effects on…”, etc.

Response: thanks for reviewer’s suggestions. The errors were corrected and the English of this manuscript was re-edited by Editage (www.editage.com). 

Response: thanks for Editor’s suggestions. We checked and revised our manuscript to meet the PLOS ONE's style requirements 

Response: thanks for Editor’s suggestions. The manuscript was polished by Editage (www.editage.com), and the language usage, spelling, and grammar were revised thoroughly. The Editing Certificate was attached below. 

“This study was supported by the National Natural Science Foundation of China (Grant Nos. 32271580, 42020104009, and 32200083). Zhejiang Provincial Natural Science Foundation (Grant Nos. LQ23D060002 and LTGS23C010001).”

Response: the role the funders took in this study was stated in the section of Founding, 

“ This research was supported by the National Natural Science Foundation of China (Grant Nos. 32271580, 42020104009, and 32200083). Zhejiang Provincial Natural Science Foundation (Grant Nos. LQ23D060002 and LTGS23C010001).”

“This study was supported by the National Natural Science Foundation of China (Grant Nos. 32271580, 42020104009, and 32200083). Zhejiang Provincial Natural Science Foundation (Grant Nos. LQ23D060002 and LTGS23C010001).”

Response: the founding information in Acknowledgments Section was deleted. 

Response: thanks for Editor’s suggestion. We accept that we will provide the relevant accession numbers or DOIs necessary to access our data if our manuscript be accepted for publication.

6. We are unable to open your Supporting Information file [Supporting information.rar]. Please kindly revise as necessary and re-upload.

Response: we are sorry for that. We previously compressed our Supporting Information file using RAR format, and we will provide the Supporting Information files using ZIP format.

---

## [Decision Letter · Decision Letter 1]

16 Oct 2023

Metabolic profiling of Mytilus coruscus mantle in response of shell repairing under acute acidification

PONE-D-23-22159R1

Dear Dr. Liao,

We’re pleased to inform you that your manuscript has been judged scientifically suitable for publication and will be formally accepted for publication once it meets all outstanding technical requirements.

Kind regards,

Amitava Mukherjee, ME, Ph.D.

Academic Editor

PLOS ONE

Additional Editor Comments (optional):

Reviewers' comments:

Reviewer's Responses to Questions

**Comments to the Author**

1. If the authors have adequately addressed your comments raised in a previous round of review and you feel that this manuscript is now acceptable for publication, you may indicate that here to bypass the “Comments to the Author” section, enter your conflict of interest statement in the “Confidential to Editor” section, and submit your "Accept" recommendation.

Reviewer #1: All comments have been addressed

2. Is the manuscript technically sound, and do the data support the conclusions?

Reviewer #1: Yes

3. Has the statistical analysis been performed appropriately and rigorously? 

Reviewer #1: Yes

4. Have the authors made all data underlying the findings in their manuscript fully available?

Reviewer #1: Yes

5. Is the manuscript presented in an intelligible fashion and written in standard English?

Reviewer #1: Yes

6. Review Comments to the Author

Reviewer #1: All my questions have been successfully addressed by authors and I believe the MS can be accepted now.

7. PLOS authors have the option to publish the peer review history of their article (what does this mean?). If published, this will include your full peer review and any attached files.

Reviewer #1: No

---

## [Editor Report · Acceptance letter]

19 Oct 2023

PONE-D-23-22159R1 

Metabolic profiling of *Mytilus coruscus* mantle in response of shell repairing under acute acidification 

Dear Dr. Liao:

I'm pleased to inform you that your manuscript has been deemed suitable for publication in PLOS ONE. Congratulations! Your manuscript is now with our production department. 

Kind regards, 

on behalf of

Professor Dr. Amitava Mukherjee 

Academic Editor

PLOS ONE